# Intracerebral Electrophysiological Recordings to Understand the Neural Basis of Human Face Recognition

**DOI:** 10.3390/brainsci13020354

**Published:** 2023-02-18

**Authors:** Bruno Rossion, Corentin Jacques, Jacques Jonas

**Affiliations:** 1CNRS, CRAN, Université de Lorraine, F-54000 Nancy, France; 2Service de Neurologie, Université de Lorraine, CHRU-Nancy, F-54000 Nancy, France; 3Psychological Sciences Research Institute (IPSY), Université Catholique de Louvain (UCLouvain), 1348 Louvain-la-Neuve, Belgium

**Keywords:** human face recognition, categorization, fusiform gyrus, face identity, SEEG, direct electrical brain stimulation, prosopagnosia

## Abstract

Understanding how the human brain recognizes faces is a primary scientific goal in cognitive neuroscience. Given the limitations of the monkey model of human face recognition, a key approach in this endeavor is the recording of electrophysiological activity with electrodes implanted inside the brain of human epileptic patients. However, this approach faces a number of challenges that must be overcome for meaningful scientific knowledge to emerge. Here we synthesize a 10 year research program combining the recording of intracerebral activity (StereoElectroEncephaloGraphy, SEEG) in the ventral occipito-temporal cortex (VOTC) of large samples of participants and fast periodic visual stimulation (FPVS), to objectively define, quantify, and characterize the neural basis of human face recognition. These large-scale studies reconcile the wide distribution of neural face recognition activity with its (right) hemispheric and regional specialization and extend face-selectivity to anterior regions of the VOTC, including the ventral anterior temporal lobe (VATL) typically affected by magnetic susceptibility artifacts in functional magnetic resonance imaging (fMRI). Clear spatial dissociations in category-selectivity between faces and other meaningful stimuli such as landmarks (houses, medial VOTC regions) or written words (left lateralized VOTC) are found, confirming and extending neuroimaging observations while supporting the validity of the clinical population tested to inform about normal brain function. The recognition of face identity – arguably the ultimate form of recognition for the human brain – beyond mere differences in physical features is essentially supported by selective populations of neurons in the right inferior occipital gyrus and the lateral portion of the middle and anterior fusiform gyrus. In addition, low-frequency and high-frequency broadband iEEG signals of face recognition appear to be largely concordant in the human association cortex. We conclude by outlining the challenges of this research program to understand the neural basis of human face recognition in the next 10 years.

## 1. Introduction: Human Face Recognition Investigated with Intracerebral Recordings

A fundamental function of the central nervous system is to recognize stimuli in their physical and social environment. Among all sensory stimuli encountered in the human environment, faces of conspecifics have a particularly important status, for a number of reasons: (1) they are biological stimuli: faces exist naturally in the animal world in many different forms [1,2]; (2) they are overwhelmingly present in the human environment, either as physical 3D stimuli in real life or videos, or as 2D images; (3) they are extremely important in the human species for social interactions; (4) unlike simple stimuli, they are not recognized based on a single dimension/property (e.g., the orientation of a grating) but through a very rich combination of multidimensional cues involving shape, texture and color properties [1,3,4]; (5) in humans, faces are recognized in many different ways and/or levels, e.g., the recognition of a visual stimulus as a face (‘generic face recognition’), of its specific identity (‘face identity recognition’) or its emotional expression (‘emotional facial expression recognition’), etc..; (6) our ability to recognize faces does not require explicit/artificial training (as for visual letter/word recognition for instance): it is a natural ability which, nevertheless, requires, and is shaped by, extensive experience during a long developmental period (e.g., [5,6,7,8]); (7) although it is an old evolutionary function and the face is used in many animal species to interact socially, there is a substantial degree of human specificity in face recognition [9,10]; (8) there is also a substantial amount of interindividual variability among human adults in their ability to perform this function, especially for face identity recognition (the most challenging face recognition function): while some people are very good at it (“super-recognizers”; [11,12]), others are not good at all (“developmental/congenital prosopagnosia”, e.g., [13]; more correctly defined as “prosopdysgnosia”, [14]). Thus, face recognition may offer a window to understand how people differ in (cognitive) brain functions in general [15]; (9) Finally, despite this variability, overall, the human adult brain performs remarkably well and is also extremely fast at various face recognition functions (e.g., [16,17,18,19,20,21,22,23]). For all these reasons, face recognition may well be *the ultimate recognition function of the human brain*, a function whose understanding should be a primary scientific objective in cognitive neuroscience research.

Human Face Recognition is defined here as the production of specific (behavioral or neural) responses to faces, these discriminant responses being reliable/reproducible across a wide variety of inputs. This general biological definition of ‘recognition’ is different than the typical definition used in experimental psychology (‘a judgment of previous occurrence’; APA dictionary of Psychology; see also [24]), with ‘face recognition’ usually used to refer specifically to ‘face identity recognition.’ Here, our definition does not imply previous experience (i.e., there could be innate recognition of a face template, for instance) and encompasses the recognition of a visual stimulus as a face (“generic face recognition”), of its specific identity (“face identity recognition”), gender (“face gender recognition”) or its emotional expression (“emotional facial expression recognition”) for instance. In this theoretical framework, since recognition responses by the nervous system need to generalize across a wide variety of inputs, ‘Recognition of faces’ is essentially the same, conceptually, as ‘Categorization of faces’ [9,25].

A key approach to understanding how the human brain recognizes faces is the recording of electrophysiological activity with intracranial electrodes (intracranial EEG; iEEG). This invasive method is used only for clinical reasons, usually in patients with epileptic seizures refractory to medication [26,27]. In these patients, intracranial recordings are performed for a few days/weeks to better define the source(s) and extent of the seizures as well as the functions of the implanted regions. Indeed, the patients are a candidate for cortical surgery to remove the brain tissue responsible for epileptic seizures while preserving as much as possible their cognitive brain functions as.

From a research perspective, the clinical procedure provides a unique opportunity to understand how the implanted brain regions contribute to recognition, in particular, face recognition. In practice, they are two surgical techniques for intracranial recordings (Figure 1). On the one hand, Electro-CorticoGraphy (ECoG) [28,29] consists in applying electrodes onto the cortical surface after removing part of the skull (i.e., subdural electrodes). Subdural electrodes have a circular shape and are spatially arranged as grids or strips with typically 5 to 10 mm inter-electrode spacing (center-to-center). On the other hand, Stereo-ElectroEncephaloGraphy (SEEG) [30,31] consists in inserting electrodes inside the brain, from the cortical surface to the medial cortex or medial temporal lobe structures (i.e., intracerebral electrodes, also often referred to as depth electrodes). The technique is called “Stereotactic” EEG because the implantation is based on a 3D coordinate system (the stereotactic frame), originally developed by Jean Talairach in the 1950s in Paris [32,33]. 

From the point of view of fundamental research, ECoG offers more extensive spatial coverage, perhaps more homogenous across individual brains. However, ECoG electrodes are limited to the gyral surface, providing a more limited sensitivity to sulcal neural activity. In contrast, SEEG penetrates the brain tissue and provides recordings directly inside the brain (i.e., not only intracranially, but intracerebrally), allowing to explore medial temporal lobe structures (e.g., amygdala, hippocampus; Figure 1B) and cortical sulci (which is particularly important given the high degree of cortical folding of the human brain, especially in association cortex [36]) (Figure 1C). The current intracerebral electrodes are thin cylinders (e.g., 0.8 mm diameter) typically containing 8–15 contiguous individual recordings sites (or contacts) of 2 mm length, separated by an insulating material (3.5 mm spacing, center-to-center).

Despite growing interest, in particular over the last decade [26,27], human intracranial recording, in particular with SEEG, remains a rare and unconventional method in cognitive neuroscience, which is often misunderstood. What can iEEG, in particular SEEG, bring to the study of the neural basis of human face recognition? Compared to other approaches, such as functional magnetic resonance imaging (fMRI), it has a number of advantages and weaknesses (Table 1).

The present review synthetizes a research program with intracerebral recordings, i.e., SEEG with depth electrodes as in Figure 1B,C) and frequency-tagging initiated 10 years ago (first recording in December 2012) that circumvent (most of) these weaknesses (Table 1) in order to provide meaningful information about the neural basis of human face recognition. The main features of the research program are the following:(1)Studies are performed in large groups of implanted patients. In mapping studies of face recognition described below, all patients with at least one electrode in the ventral occipito-temporal cortex (VOTC, including the occipital lobe (OCC), Posterior Temporal Lobe (PTL), and Anterior Temporal Lobe (ATL) (Figure 1C)) are included, with samples of up to, e.g., N = 121 patients for some studies [37].(2)Patients are well characterized in neuropsychological functions, in particular regarding their ability to recognize face identity [38]. Unfortunately, this critical issue is neglected most of the time in iEEG studies of face recognition or other cognitive functions in general.(3)The use of face and nonface stimuli that are carefully controlled to eliminate or minimize the contributions of unreliable low-level sensory features. A ‘natural’ control to isolate high-level face recognition processes (e.g., through stimulus inversion or variability of low-level sensory features in the stimuli grouped in single condition) is often preferred to artificial procedures of global normalization or elimination of visual cues that degrade visual stimuli and may paradoxically increase low-level confounds by making local cues highly salient (e.g., [39]).(4)The use of tasks that are relatively easy for the patients, with implicit measures (i.e., tasks that do not require explicit recognition of faces).(5)Finally, a key component of this research program is the presentation of stimuli at periodic (i.e., fixed frequency) rates, relatively rapidly, e.g., six images/second during stimulation sequences of about 1 min. Given the periodic stimulation mode, the brain’s electrophysiological responses to the stimulation can be measured and quantified in the frequency domain. This fast periodic visual stimulation (FPVS) approach, is also referred to as “steady-state visual evoked potentials” (SSVEP; [40]) or “frequency-tagging” [41]. However, besides referring to the type of response rather than the approach, SSVEP is a loaded term, which suggests that there is an inherent difference between SSVEP and standard event-related potentials (ERPs) (e.g., see [42]). In reality, an SSVEP is no more than an ERP evoked periodically and expressed in the frequency-domain. When the temporal distance between periodically evoked ERPs is too small, they tend to overlap to the point where the response may appear as an externally-evoked “oscillation” [43]. This creates substantial confusion, with the frequent claim that SSVEP is an oscillatory brain response (with the same frequency as the flickering stimulus) that results from “entrainment” of spontaneous EEG oscillations (e.g., [44,45,46,47,48,49]). This interpretation is speculative and not necessary. The most parsimonious account of the fast periodic responses (i.e., superimposition with or without temporal overlap of ERPs; [43,50,51] is preferred here). Regardless, this approach provides objectivity in the identification of the response of interest (i.e., exactly at the frequency known by the experimenter) and its quantification (amplitude in the frequency-domain exactly at the stimulation frequency). The technique is also associated with high sensitivity (i.e., high signal-to-noise ratio, SNR), which is important even when recording large responses inside the brain (as compared to weaker responses in scalp recordings) (see [52,53] for reviews).

## 2. A Cartography of Human Face Recognition with SEEG

A primary cartography of human face recognition is based on a simple paradigm in which natural images of biological stimuli (animals, plants, etc.) and man-made objects appear at a fast rate of 6 Hz (6 images by second), with highly variable natural images of faces inserted every five stimuli (i.e., 1.2 Hz) (Figure 2).

This simple paradigm was first reported in 2015 in scalp EEG [54] and has been validated and used in more than 25 scalps EEG studies to date (e.g., [43,56,57]) as well as in MEG [58]. The paradigm is based on the following principle: while the common neural activity to faces and nonface stimuli is projected to the 6 Hz base rate, neural activity elicited in response to faces that systematically differs from neural activity to nonface objects would appear in the EEG spectrum at 1.2 Hz (and specific harmonics, i.e., 2.4 Hz, 3.6 Hz, etc.; [54,59]; Figure 2C). Thus, and this is an important point, neural activity recorded at 1.2 Hz, which corresponds to the rate at which faces are presented in the sequence, does not merely reflect face-evoked or face-related activity but face-selective activity: given that faces are presented inside a train of periodically presented nonface stimuli, if they do not elicit any specific response, there is no activity at 1.2 Hz. Importantly, this selective activity must also be sufficiently constant across different exemplars of faces appearing in the sequence (usually about 50 face exemplars varying in viewing conditions; [54,60]) to be detectable in the EEG spectrum. Therefore, the paradigm allows identification and quantification of selective (i.e., differential) neural responses to faces (at 1.2 Hz) that generalize across widely variable stimuli: a (generic) face recognition function for the human brain. Importantly, since these recordings are quantified in the frequency domain following a Fourier analysis, the common neural activity to faces and objects which is tagged at 6 Hz is naturally separated (or filtered out) from the face-selective response at 1.2 Hz. 

Figure 2C illustrates a typical EEG frequency spectrum (expressed in signal-to-noise ratio, SNR) for a recording contact at the border of the right fusiform gyrus (FG) and occipito-temporal sulcus (OTS) in a single brain (2 periodic stimulation sequences). The OTS region is included in the region-of-interest of the lateral fusiform gyrus (see below). In Figure 2C, note the clear peaks in the EEG spectrum exactly at 6 Hz (and harmonics, here the second harmonic at 12 Hz), which again reflect the common neural activity to faces and nonface stimuli. Since the whole 60 s of SEEG recording during stimulation is Fourier transformed, the spectrum has a very high-frequency resolution (i.e., 1/60 = 0.0166 Hz). This high-frequency resolution not only allow unambiguous identification of the narrow peaks related to the stimulation but—together with the high number of contrasting stimuli presented in a short amount of time—contributes to the high SNR: while the broadband noise is distributed in many frequency bins, the signal concentrates in a narrow bin contaminated by little of the noise [61,62]. 

Critically, when the faces appear periodically, additional peaks at 1.2 Hz and their specific harmonics (2.4 Hz, 3.6 Hz, etc.) are present in the SEEG spectrum. These peaks are not present if the same face stimuli do not appear periodically in the stimulation sequence (lower spectrum in Figure 2C). Moreover, note that sometimes the peak is the highest at the second or third harmonic, for instance, rather than at the fundamental/first harmonic frequency of 1.2 Hz. These harmonics reflect the fact that the differential response elicited by the faces in the time-domain is not a pure 1.2 Hz sinewave (in fact, it is a series of peaks and throughs, as a sequence of differential event-related potentials specifically evoked by the faces; see, e.g., [43,54,56,59,63]. Thus, importantly, with this simple approach, there is no need to invoke the generation of “brain oscillations” or the “entrainment of spontaneous brain oscillations” by the periodic stimulation: simply, differential (i.e., selective) responses to faces are captured at a (1.2 Hz) periodic rate (defined by the experimenter), and these differential neural responses are expressed in the frequency-domain (i.e., a SSVEP is a periodic ERP expressed in the frequency-domain, see above).

Once the EEG frequency spectrum has been computed, the generic face recognition response can be quantified by summing the (baseline-corrected) amplitude at significant harmonics (see [43,59] for justification of the summation procedure for quantification). In SEEG, this is performed for each recording contact, which can then be evaluated for statistical significance by using the mean and standard deviation of neighboring bins of the frequencies of interest [41,62,64] (Figure 3).

This procedure provides, for instance, a large number of significant contacts, as shown on the right of Figure 3, which is a map based on a normalization of each individual brain in a standard anatomical atlas (Talairach). In our first report, across 28 individual brains, we found 555 contacts, or 23% of the grey matter contacts in the VOTC, with a significant face-selective response [60]. This proportion of contacts has remained relatively constant with larger samples (e.g., ~27% for N = 84; Figure 3). This analysis of a large-scale exploration provides us with a first global picture of the contribution of the VOTC to human face recognition with SEEG [60].

At first glance, the wide spatial distribution of face-selective activity as found with SEEG recordings contrasts with functional magnetic resonance imaging (fMRI) findings of (apparently) spatially isolated and well-defined face-selective regions in the VOTC, most prominently the so-called Fusiform Face Area (FFA), in the lateral section of the middle fusiform gyrus (for reviews on these fMRI face-selective regions in the human brain, see, e.g., [64,65,66]; [67]; Figure 4).

Given the difference observed, it is legitimate to ask which of the two techniques provides the correct representation of face-selective activity in the human VOTC. SEEG (or ECoG) records electrical fields directly linked to neural activity. These electrical fields spread in the brain tissue and can be measured at a distance from the actual neural source of the field. This distance, and therefore the ‘localness’ of the measured neural responses, depends on a number of factors that relate both the local anatomy and physiology (see, e.g., [68]) and the recording and analyses methods (e.g., the size and material of the electrode, the choice of the ‘reference’ electrode used to determine electrical potential). In SEEG or ECoG recordings, in most cases, the neural activity recorded is, therefore, not only (although predominantly) local and may originate from neural sources at a distance (up to a few millimeters from the electrode) from the recording site. This may potentially lead to an inflation of the proportion of face-selective contacts. Moreover, the apparent wide distribution highlighted in SEEG (as seen in Figure 3 and Figure 4) is partly driven by the display of group data rather than individual maps. Yet, it is also possible that, even at the individual subject level, an arbitrary statistical threshold in an fMRI face localizer leads to the identification of only the tip(s) of an iceberg, i.e., a widely distributed and continuous face-selective neural activity in the VOTC.

In order to help clarify this issue, face-selectivity in different cortical regions can be quantified using the (noise-corrected) amplitude of the response recorded at each contact. Recording contacts are grouped in anatomical regions-of-interests (ROIs), by labeling each recording contact in each individual brain based on precise anatomical criteria, as illustrated in Figure 5A. This quantification reveals differences in face-selective amplitude across different ROIs, with one region, in particular, showing a much larger response than the others: the lateral section of the fusiform gyrus (latFG) in the right hemisphere (Figure 5B). 

In addition, at the individual level, even though a single participant can display a large number of significantly face-selective contacts, there is a wide variability in face-selective amplitude across contacts, with a clustering of the most face-selective contacts (i.e., highest amplitude) around consistent anatomical regions ([60]; Figure 6A). This variability in face-selective amplitude across the VOTC can also be appreciated at the group level by spatially smoothing the amplitudes computed in each contact transformed in the Talairach space, revealing the differences in face-selective amplitudes across the VOTC (Figure 6B). On such maps, one can appreciate that the largest amplitudes concern a bilateral strip of cortex from the lateral inferior occipital gyrus extending up to the temporal pole along the fusiform gyrus and surrounding sulci. There is a clear right hemispheric advantage, especially at the level of the middle section of the lateral fusiform gyrus, with a peak of activity that is extremely close—only slightly anterior—to the FFA as defined in fMRI (Figure 4). Hence, when considering amplitude maps, the discrepancy between SEEG and fMRI is no longer striking.

## 3. Putting Findings into Context

The findings summarized above were originally reported in the paper of Jonas and colleagues in 2016 with N = 28 individual brains explored, starting just about 10 years ago (first recording in December 2012). However, this publication was preceded by a large amount of research on the neural basis of human face recognition performed for decades, including neuroimaging and intracranial recordings with ECoG (mainly) and also SEEG. Thus, it is important to acknowledge what is *not* novel in these observations and put them in context. Indeed, several studies performed in the 1990s already described face-selective human intracranial (ECoG) activity in the VOTC of the population of epileptic patients (e.g., [70,71,72,73]). These original studies appeared at about the same time as metabolic neuroimaging studies reporting face-selective activity, first with positron emission tomography (PET; [74]) and then with fMRI [67,75,76,77]. Even with SEEG, evoked potentials in response to faces had been described early on in the VOTC and other brain regions, although with little or no measure of category-selectivity ([78]; see also [79]). Although intracranial investigations of human face recognition remain rare, there have been further studies over the last three decades with various paradigms and stimuli, although usually with lower samples of patients (e.g., [80,81,82,83]). These face-selective responses have been primarily found in the ventral surface of the brain, in various regions of the VOTC, as illustrated above. They have been described in terms of phase-locked low-frequency activity in the time-domain (i.e., face-selective potentials such as the N200; [70]) or more recently broadband activity most visible in the gamma frequency range (i.e., high frequencies; ‘broadband gamma range’; [84,85,86,87]) or both [37,80,81,83,88].

Nevertheless, with respect to these studies, thanks to the objective quantification in the frequency domain in our approach, a number of original observations were made [60]. First, to our knowledge, the well-known right hemispheric (RH) advantage of human face recognition, described with multiple methods and specific to the human species (see [89,90]), had never been reported in human intracranial recordings before the study of Jonas et al. (2016) [60] [Note that Allison et al. (1999, [71]) did report a significantly larger estimated width of face activations over the right compared to the left hemisphere, despite finding no interhemispheric difference in the N200 amplitudes or in the proportion of face-selective (relative to recorded) electrodes]. Second, previous intracranial studies did not provide quantitative cartography of the function across the whole VOTC. Here, this cartography, as illustrated in Figure 6, emerges from the objective quantification of the whole face-selective response in the frequency-domain. Third, thanks to this wide-scale cartography, the study of Jonas et al. (2016) provided the first evidence with direct recordings of brain activity that the lateral portion of the middle fusiform gyrus in the right hemisphere (i.e., the location of the right FFA) shows the largest face-selective amplitude across the whole VOTC (Figure 6). Indeed, before that study, even though ECoG studies showed good correspondence between face-selective electrophysiological activity and BOLD signal in the fusiform region [71,88,91,92], general cartography had only been performed in metabolic neuroimaging studies (in many fMRI studies starting with [67,77]; for evidence of the FFA in PET see [93]). However, for all of its virtues, fMRI also suffers serious limitations, for instance, large fluctuations of SNR across brain regions due to proximity to veins and magnetic properties of the tissue [94,95]). In particular, there is a well-known large magnetic susceptibility artifact anteriorly to the typical location of the FFA in the ventral anterior temporal lobe (VATL) [95,96]. Hence, face-selective activity in this region is virtually invisible in fMRI or at least seriously underestimated (even with distortion-corrected fMRI sequences, as discussed below) [96,97]. This is not the case with intracranial recordings, especially SEEG recordings that can be performed also in cortical regions that are not superficially exposed (i.e., sulci), and constitutes about 70% of the cortical surface of a human brain on average [36]. Thus, the finding that the peak of face-selective activity in the human VOTC as determined in SEEG falls at a location in the lateral section of the mid-fusiform gyrus that is highly similar as—only just slightly anterior—to the FFA as defined in fMRI studies (Figure 7) is truly remarkable. Moreover, our SEEG study showed that the largest face-selective response was located in the lateral rather than the medial section of the fusiform gyrus (Figure 7), in line with fMRI findings, as particularly emphasized by Kevin Weiner, Kalanit Grill-Spector and their colleagues [98,99] (see also [71,88] in ECoG). For these reasons already, the importance of these intracerebral findings, therefore, go beyond the issue of the neural basis of human face recognition: it supports the view that the population studied, i.e., epileptic patients, is valid to understand the neural basis of a complex, distributed, cognitive function. In addition, if needed, it also strengthens the validity of the frequency-tagging paradigm used.

Fourth, another key finding of this original approach was the extension of face-selective activity along the ventral anterior temporal lobe (VATL) (Figure 6). Specifically, these face-selective responses are consistently observed in the anterior fusiform gyrus, anterior occipito-temporal sulcus, and anterior collateral sulcus. Responses in these regions have not been highlighted with fMRI due to the above-mentioned large magnetic susceptibility artifact in the VATL, anteriorly to the typical location of the FFA. Although face-selective activity had been reported previously in the VATL with ECoG [73], it is rarely the case, and, again, no systematic mapping has been provided. Moreover, again, with fMRI, the large magnetic susceptibility in this region generally precludes finding face-selective activity, except very anteriorly in the temporal pole. Hence, there is generally a gap between face-selective fMRI activity in the region of the lateral middle fusiform gyrus (FFA) and the temporal pole (e.g., [100,101,102,103]) (Figure 7). Our study fills this gap and contributes to clearly highlighting VATL as an essential part of the brain network involved in human face recognition.

**Figure 7 brainsci-13-00354-f007:**
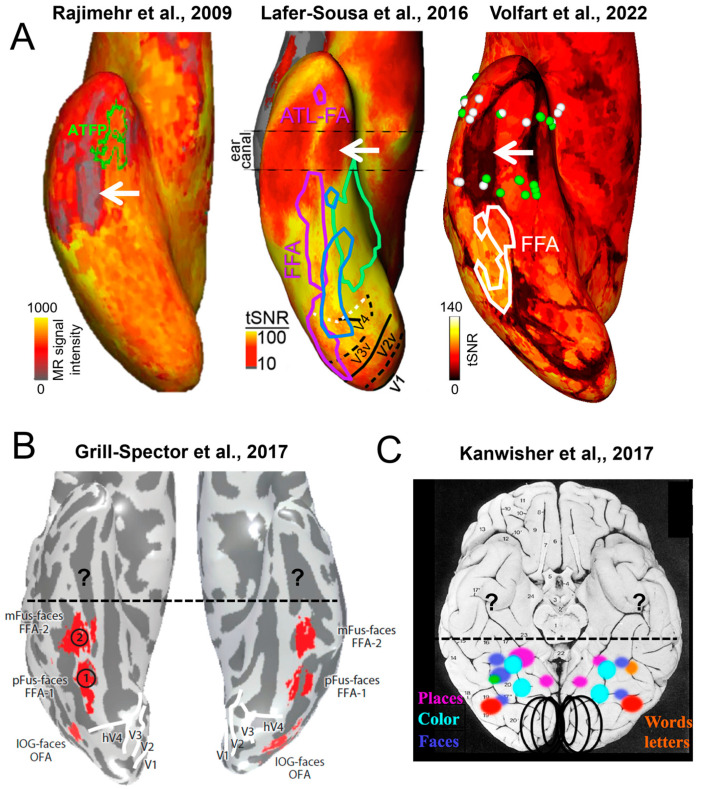
Magnetic susceptibility artifact due to the ear canal (or vATL signal dropout) in fMRI and its consequences on the current picture of fMRI neural basis of human face recognition. (**A**) These examples show the BOLD signal level and tSNR on ventral views of the inflated cortical surface. The vATL signal dropout is indicated by a white arrow (gray or black zone in Rajimehr et al., 2009 [101]; and in Volfart et al., 2022 [97]; red zone in Lafer-Sousa et al., 2016 [104]; reproduced with permission). In these regions, BOLD measurements are not reliable. Note that face-selective activations (green outlines in Rajimehr et al., 2009, purple in Lafer-Sousa et al., 2016, and white in Volfart et al., 2022) are found all along the VOTC except in the region of the signal dropout. On the contrary, SEEG, which is not affected by SNR variations, is still able to capture face-selective response in this region (green dots in Volfart et al., 2022). (**B**) Topological organization of the face-selective activations in the VOTC (red spots) in one of the most recent reviews on the functional architecture of face perception in the VOTC (Grill-Spector et al., 2017, [66]). Note that there are no activations anteriorly to the middle fusiform gyrus (FFA, spot indicated by “2” in the figure). (**C**) Schematic representation (from Kanwisher, 2017 [67]) of the progress made in the understanding of the functional architecture of the VOTC between 1990 (birth of fMRI) and 2017. Although the posterior VOTC is refined (see colored spots representing schematic category-selective regions), little or no progress appears to have been made in mapping anterior VOTC regions with fMRI.

Finally, our frequency-tagging investigation also revealed an interesting phenomenon: at some recording contacts showing face-selective peaks, there was no response at all at 6 Hz (or its specific harmonics, 12 Hz, etc.) (Figure 8). It is as if the local populations of neurons responded exclusively to faces. Importantly, we found that the proportion of contact showing such face-exclusive responses increased from posterior to anterior regions, reaching up to 40% of the contacts in some VATL regions (anterior collateral sulcus, aCoS; anterior occipito-temporal sulcus, aOTS). This is interesting because each contact records the activity of hundreds of thousands, if not millions, of neurons (see [105] for neuronal density in the human fusiform gyrus), suggesting that exclusive responses to faces may exist even at this relatively coarse level of spatial resolution.

The data reported by Jonas et al., 2016 (Available at: https://datadryad.org/stash/dataset/doi:10.5061/dryad.5f9v7, accessed on 17 February 2023) was based on N = 28 implanted patients. This was already quite a large number, but our current sample tested with the same paradigm is 121 patients [37]. Remarkably, the main findings, as described above, have all been confirmed, although the cartography has been extended and refined.

## 4. Implications for Understanding Face Recognition

What can we learn from studies such as the one that just described that map face-selective activity in the human brain? Why is there an obsession with face-selective activity and its localization in the human brain after all (as evidenced by the large number of neuroimaging studies that rely on so-called ‘face-localizers’ in fMRI; [67])? In the 1980s–1990s, when functional neuroimaging emerged, many psychologists and cognitive scientists considered that localization of function in the brain would not help us understand these functions in any way (e.g., [106]). Nowadays, this point of view is certainly not prevalent in the scientific community, but there are still some fierce opposition to what some merely see as a phrenology approach to brain function [107]. Therefore, it is important to reflect on the implications of our observations.

At the outset, we have defined face recognition as the production of specific responses to faces, responses that are reliable/reproducible across a wide variety of facial inputs. This is the problem that we are dealing with when having to recognize faces in the environment: our central nervous system has to provide a response that discriminates these faces from other stimuli and is able to do so under a wide variety of viewing conditions. There is nothing more than that, and thus face recognition may, in fact, appear very simple, similar to making a footprint in the sand [108]. In contrast, according to the standard way of conceptualizing this face recognition function in cognitive (neuro)science, this apparent simplicity masks a very complex chain of processes: along David Marr’s (1982; [109]) theory for visual recognition, the mind/brain extracts ‘information’ on faces and through a series of hierarchically organized computational processes, sequentially derives visual representations in order to achieve “invariant” face recognition [66,110,111,112]. However, it is important to emphasize that defining these representational/computational stages is not the scientific problem that we are dealing with but is only one way of conceptualizing this problem, i.e., into a computational framework. This framework may be correct, but it may also be fundamentally wrong, and we can certainly conceptualize the problem of human face recognition very differently.

In any case, by definition, face-selective responses in the brain are important since they just reflect the very function that we are trying to understand. As soon as there is (sufficient) face-selective activity, the system can be said to have recognized the stimuli as faces. Therefore, in line with a distributed cortical model of memory (‘Systems Memory’; [113]), we consider that these face-selective activities in the VOTC reflect our cortical memories of faces. Moreover, unless one adopts a nativist perspective and considers that, at birth, the populations of neurons involved were already selective to faces (or that these populations of neurons were predetermined genetically to become selective to faces at some point during development, regardless of experience), these cortical memories of faces in the VOTC must have emerged due to extensive experience with faces in the human environment from early in life (e.g., [114]) throughout development and lifetime 

We think that finding where these memories of faces are located in the brain, that they are spatially distributed from posterior to anterior VOTC regions but not fully and uniformly distributed (as in connectionist computer simulations, for instance, with “representations” distributed across all “synaptic weights,” or across a continuous space as suggested by some fMRI decoding studies, e.g., [115,116]), that they tend to form clusters with local peaks of activity, and that there is an overall posteroanterior increase in abstraction (i.e., relatively more face-selective responses in posterior sites and face-exclusive responses in anterior sites) constrains and contributes significantly to our understanding of how the brain achieves the face recognition function (and recognition in general).

From this point of view, functional mapping of the cortex is important, especially if the cartography concerns a well-defined function. Of course, in order to understand how the function is accomplished in the brain, we need to go further and explore connections between brain regions, the timing of neural responses across these regions, finer levels of spatial resolution, criticalness of these responses (with focal electrical stimulation), etc. Given that this research is particularly challenging to perform in the human brain, with a number of constraints (e.g., Table 1), it is, therefore, legitimate to ask why we should not rely instead on animal models to answer these questions, in particular non-human primates. Let us make a parenthesis here and briefly address this issue.

## 5. Human Specificity

In the neuroscientific community, the monkey brain is widely considered the best available model to understand the human brain [117,118], especially for visual function [112,119]. In particular, there is a long tradition of research to study monkeys, usually macaque monkeys, to understand the neural mechanisms of human face recognition [120,121,122,123,124]. Studies tracing back to the 1970s and 1980s have found neurons responding selectively to faces in the monkey temporal lobe ([120,125]; see [121] for a review of early studies), and this species has been studied intensely over the last two decades by a combination of fMRI and electrophysiological recording methods in a few laboratories in the world (e.g., [110,126,127,128] see [123,124] for reviews). FMRI studies have found a number of cortical face-selective regions in the monkey temporal lobe, mainly in the superior temporal sulcus (STS; e.g., [129,130,131,132,133]), with high proportions of face-selective neurons in these regions [128,134]. This research program has led to bold proposals regarding how face identity is coded in the human brain ([135]; see also [136,137]) and how a whole network of brain regions interacts to recognize faces in general ([123,124] for reviews).

However, while such monkey studies can potentially bring important information to understand human face recognition mechanisms, they should also be taken with great caution, as they are severely limited at three levels least. First, monkeys’ face recognition abilities are not well documented and, in comparison with humans, appear to be severely limited. This is especially the case for face identity recognition: monkeys’ natural ability at this function is poor, they need hundreds of trials to reach modest performance at simple discrimination tasks, and they appear to rely on qualitatively different processes than humans’ [9,10,138,139,140] (see also, e.g., [141,142] for limitations at recognizing the gender of conspecific monkeys from their face only for instance). Second, the face-selective regions defined here in the human VOTC with intracerebral recordings do not exist in the monkey brain. Macaque monkeys have a brain that is about 13–15 times smaller in size and number of neurons than a human brain [143] (Figure 9), with a relatively much smaller temporal lobe that is also quite different than the human temporal lobe in gyrification [144,145]. In particular, the fusiform gyrus is a hominoid structure, identifiable in the ape brain but not the monkey brain ([99,146,147]; Figure 9). Third, in contrast to humans, for whom right hemispheric lateralization of face recognition is well established (see the review of [90]), as also shown above with intracerebral recordings and critically linked to function, there is no evidence of such (population-level) lateralization in the monkey brain for face stimuli (e.g., [130]).

Despite these substantial neuroanatomical differences, neuroscientists generally consider that the cortical face network of the macaque monkey—essentially located in the STS of that species—is largely homologous to the human cortical face network, i.e., that these systems would originate from a common primate ancestor ([130,148]; see also [103,149]). According to this view, the monkey cortical face network would be a sort of “mini-model”, already present in the last common ancestor of humans and monkeys, which would have been largely preserved, although considerably enlarged in size in humans, during 25–30 million years of primate evolution. However, this far-fetched view of neuroanatomical-functional homology between the two species’ cortical face systems is not well supported (see [9,10] for detailed arguments). While we can present monkeys with thousands of images and undoubtedly elicit face-selective activity in their brain, at the system level of organization, the regions identified may have little to do with those found in humans in the VOTC and tell us little about how the human brain recognizes faces, in particular for face identity recognition [9,10,150]. For this reason, even considering the limitations of this research (Table 1), we think it is more fruitful to further explore our own species’ brains with intracerebral recordings as described above to understand human face recognition.

## 6. Contrasting Categories

### 6.1. Face-Selective vs. House-Selective Responses

Using the same SEEG approach as described above, we contrasted spatial cartographies of human face-selective activity in the VOTC to category-selective response to pictures of houses. That is, we presented a very large sample of patients (N = 75) with stimulation sequences in which, instead of faces every five stimuli, pictures of houses were included (Figure 10). In that study, a large variable set of houses (Figure 10A) were used in the stimulation sequences (Figure 10B). Why contrasting pictures of houses with faces? First, this comparison provides a means to further validate the FPVS-SEEG approach in epileptic patients: if we find the same category-selective regions for houses as for faces, it will show that our approach is not valid; i.e., that it merely measures a generic category-selective response in the human brain, maybe due to the periodic (1/5) repetition of any object class. Second, a spatial dissociation between face- and house-selective neural activity would further validate the approach, especially if the house-selective activity is found in similar regions as in fMRI, i.e., in more medial regions of the temporal lobe such as the parahippocampal gyrus and Collateral Sulcus [151]. These responses to houses are not thought to reflect category-specific processes for houses, but these types of stimuli typically serve as spatial landmarks (at least in the modern world) and therefore appear to recruit those regions of the brain involved in spatial orientation. 

This study reported a number of interesting findings [152]: (1) clear recording contacts with face-selective responses and no house-selective response, or the opposite (Figure 10C); (2) a much larger (about three times) proportion of contacts showing only face-selective responses as compared to house-selective only contacts, in line with findings that the category-selective response on the scalp of neurotypical human adults is much larger for faces than houses with this paradigm [63]; (3) a right hemispheric dominance for faces but not for houses, also in line with scalp EEG and neuroimaging studies; (4) a clear spatial dissociation of face-selective and house-selective contacts with a lateral-medial distinction, respectively (Figure 11A), replicating fMRI observations, but also intracranial (EcoG) studies with smaller samples ([85,88]; see also [153]); (5) An original extension of this dissociation to anterior temporal regions of the VATL (Figure 11A).

Two additional findings are worth mentioning. First, there are much fewer (i.e., four times) house-exclusive contacts than face-exclusive contacts, as shown in Figure 8, in particular in the VATL. Moreover, there is almost no overlap between these category-exclusive contacts, indicating that the face-exclusive contacts are not just domain-general responses to “predictable” events (e.g., “oddball faces” every five stimuli) among nonface objects. Second, considering category-selective responses, there is still substantial overlap (37.7% of contacts) between faces and houses (Figure 11B; see Figure 10C for an example of contact). That is, 37.7% of contacts show both face-selective and house-selective activity (i.e., a differential response as compared to all other visual object categories). There are two potential accounts for such findings. On the one hand, the very same population of neurons may be involved in responding selectively to both faces and houses. On the other hand, these contacts may record spatially distinct but close populations of neurons. Both the respective amplitude maps of these responses, as well as the lack of correlation of amplitude – despite near-to-ceiling correlation for the base rate responses - across these contacts support the latter view (Figure 11C).

In summary, this extension of our frequency-tagging approach to map face-selective and house-selective neural responses in the human brain supports the validity of the approach—by showing a neat spatial dissociation that corresponds to neuroimaging studies in neurotypical individuals—and extends it to anterior regions of the temporal lobe (again under sampled in fMRI). It also indicates that populations of (VATL) neurons responding only to faces are not just generic oddball detectors.

### 6.2. Face-Selective vs. Word-Selective Responses

Most recently, the same approach was applied to contrast face-selective neural activity and selective activity to written words as recorded intracerebrally, this time on a sample of 37 patients. Why contrast human faces and words? At first glance, these categories of stimuli differ at many levels. Faces are 3D dynamic biological stimuli, predominately curved and made of parts that do not stand alone. In contrast, written words are artificial 2D shapes containing edges and junctions, with individual parts–letters–that may recombine for other meanings. Face recognition is acquired incidentally and improves from birth to adulthood while learning to read requires an effortful, explicit learning process starting, usually only after a few years of development. While faces convey a lot of social cues, leading to many types of recognition functions (e.g., identity, expression, gender, head orientation, etc.) and may be associated with names and both verbal and nonverbal semantic information, words are strongly linked to phonology, language production, and more generally to semantics (see [90] for discussion of these differences).

In the time-scale of the evolution of the species, visual word recognition is a recently acquired skill (about 5000 years), and until a few hundred years ago, only a small proportion of the population could read. In fairness, human face recognition has also certainly experienced significant changes in most human populations over the last 5000 years (i.e., since the emergence of early civilizations and populations of thousands of individual Homo Sapiens living in cities). Yet, it is arguably an older evolutionary skill which, unlike visual word recognition, is partly shared by other animal species (although with important differences even compared to non-human primates, as discussed above; see also [140,154] for substantial differences in face recognition between humans and chimpanzees).

Despite these large differences, similarities in processing faces and written words are often emphasized in cognitive (neuro)science, in particular, because both types of stimuli would require foveal analysis and would be composed of parts that are integrated into a unified—so-called holistic—representation (e.g., [155,156]). For these reasons, it has even been claimed that faces and written words substantially overlap in neural representation [157,158], a surprising claim because a neural ‘representation’ must have, by definition, a systematic (i.e., consistent) specific relationship with the sensory stimulus that it ‘represents’ [Note: A ‘representation’ could be simply defined as (a pattern of) activity in the neural system that has a systematic relationship with a sensory stimulus of the environment, a motor action, or another pattern of activity. Although this term has been used in the past two decades in neuroimaging in the restrictive context of multivariate pattern analysis (MVPA; [159]], neural activity does not have to be distributed across arbitrary spatial units such as voxels. That is, strictly speaking, a larger neural activity to faces than nonface stimuli in the right lateral middle fusiform gyrus as a whole constitutes a “representation” of face stimuli.). If it is not the case, one cannot discriminate, or recognize, these stimuli: a face cannot be taken for a written word or vice-versa.

Related to that idea of overlap between neural representations of faces and words is the hypothesis that the well-known right hemispheric dominance for human face recognition, which has been illustrated here with intracerebral data and is supported by a wealth of evidence, would be due to neural competition with the selective visual representation of letters and words in the LH, in particular in the left VOTC [157,158,160,161]. That is, during reading acquisition, populations of neurons in the left VOTC, essentially in the left middle fusiform gyrus, would become selective to letter strings so that selectivity to faces initially supported by both left and right homologous VOTC regions would be reduced in the LH, therefore relying increasingly on RH structures. Hence, the reading acquisition would directly cause RH lateralization for faces. In truth, this hypothesis of right lateralization for face recognition being due to neural competition in the left hemisphere with the language function is quite old [162], but it is only in relatively recent years that it has received so much attention, with a focus on reading in the VOTC. However, despite the prevalence of neural competition for sensory-motor representation in the central nervous system [163,164] and evidence of such competitive interactions between language and face recognition [165,166], there is virtually no evidence supporting the specific view that right lateralization of face recognition is caused or even increased due to competition with written word representation in the left VOTC (see [90] for an extensive review).

Nevertheless, to provide original information on this issue, we contrasted category-selective intracerebral responses to faces and written words as obtained in the same individual brains in two fast periodic stimulations paradigms (Figure 12). While faces were—as described—compared to nonface objects, written words were compared to pseudofonts as generated in previous studies by rearranging the letter strokes [167]. 

Admittedly, this comparison is slightly unbalanced because the base stimulation rates differed between conditions (6 Hz for faces among objects; 10 Hz for words among pseudofonts). Yet, within a certain range (i.e., not too fast or too slow), variations of these stimulation rates do not change at all the nature of the selective neural response [169]. It could also be argued that pseudowords (i.e., pronounceable words), rather than words, should have been used for a better comparison with unfamiliar faces (that are not associated with semantics). However, since visual words and pseudowords appear to recruit the same VOTC region(s) [170], this should not influence the conclusions.

Here are the main findings of this study [168]. First, if one excludes the contacts showing selective responses to both categories, there is a clear contrast between a bilateral/RH dominant category-selective response for faces in the VOTC and a left-lateralized category-selective response for words (Figure 12). While this is fully expected based on neuroimaging studies, it had not been shown at the level of whole VOTC maps in intracerebral recordings, again supporting the validity of the technique and the population tested. Second, in the RH, most face-selective recording contacts (80%) do not respond to visual words, showing a clear spatial dissociation between these categories. In the LH, however, about 50% of face-selective contacts also show selective responses to words (Figure 13). From the perspective of word-selective contact, some show no response at all to faces (example in Figure 13). Yet, about 63% of these contacts also show responses to faces (Figure 13). Thus, even if the overlap in category-selective response is only 30.74% of significant recording contacts overall, the overlap in representation for faces and words is substantial in the left hemisphere. Moreover, most of the face-word-overlap contacts (70%) were found within anatomical regions that have been proposed to be central to face and word recognition processes, in particular, the fusiform gyrus and surrounding sulci.

How is it possible that a recording contact shows category-selective neural activity to both faces and written words? What could be the functional meaning of such a finding? As mentioned above, each contact (of 2 mm length, Figure 1) in SEEG reflects the (postsynaptic) activity of hundreds of thousands, if not millions, of neurons that can be located at a distance (on the order of millimeters) from the recording contacts. Therefore, while this selectivity to both face and words could reflect a genuine overlap in the neural representation of faces and written words, it could be that the category-selective responses for faces and words are truly spatially dissociated but close enough to generate activity at a number of common recording contacts (due to diffusion of electrical field). How can one clarify this issue? Again, we turned to correlation measures using amplitudes at each recording contact, reasoning that if the very same neural population is involved in processing selectively for faces and words, then contact with a high (low) face-selective response amplitude should also show a high (low) word-selective amplitude. In contrast to this prediction, in the face-word-overlap contacts, a correlation between the face- and word-selective amplitudes in the left and right hemispheres was not significantly higher than 0 (Figure 14), supporting the view that these contacts truly capture different neural activities from nearby sources.

These observations go against the view of commonality of representation or active competition (which would have been reflected by negative correlations) between face-selective and word-selective neural populations in the human brain, including in the middle section of the fusiform gyrus and surrounding sulci. This conclusion generally agrees with other sources of evidence regarding this issue [90], but the findings summarized here with direct measures of activity are original and particularly compelling (Hagen et al., 2021 [168]). In addition, the spatial dissociation in the left fusiform gyrus identified in our intracerebral study points to a slightly more lateral word-selective than face-selective activity, as shown in Figure 15A,B (see details in [168]). This is in line with findings of fMRI studies that have focused on the comparison between the so-called FFA and VWFA in the fusiform gyrus (e.g., [171], as shown in Figure 15C).

## 7. Face Identity Recognition

Having focused so far on the mapping of category-selective neural responses to faces, what has been defined here as generic face recognition (or generic face categorization), we now turn to map another function in the VOTC: the recognition of someone’s identity from their face, i.e., face identity recognition (FIR). FIR is clearly the most challenging face recognition function and perhaps the most challenging recognition function at all for the human brain. There are many reasons for that (see [9]), but let us just focus on the three main ones. First, while individual faces certainly differ more in humans than in other animal species [172], all human faces, in particular within a genetically homogenous group, share similar features and their overall configuration, so FIR requires relatively fine-grained visual discrimination processes. Second, the same face identity can vary substantially under different viewing conditions to the point where different views of the same identity may often differ physically more than different facial identities [173]. Hence, FIR requires great generalization capacities in order to treat physically different stimuli as belonging to the same (category of) identity. A third factor, usually neglected, is that, in most modern human societies, the number of different people encountered (in real life or through the media) is extremely large and variable over time. For this reason, the number of categories of human facial identities is, if not infinite, extremely large and undetermined. Hence, contrary to other face recognition functions (e.g., emotional expression, sex, age, etc.), for which there is a small and relatively stable number of potential categories, FIR is a wide open flexible recognition function, making it extremely challenging [9].

Despite this challenge, in humans, identity recognition is primarily based on the face, which, among body parts, carries by far the largest morphological and genetic diversity [172]. Young adults in modern societies are able to accurately identify thousands of faces on average [174], rapidly (i.e., at a glance and within a few hundred milliseconds of processing) and largely automatically (i.e., without volition) (e.g., [17,23,175]). Thanks to these characteristics (accuracy, speed, automaticity, and large and variable number of exemplars to recognize), neurotypical human adults can be considered genuine experts at FIR [5,176].

Most of the research on human FIR, including its neural basis, is based on pictures of unfamiliar faces for good reasons: they offer better stimulus control, and the same stimuli can be presented to different participants tested. However, when human subjects have to match pictures of faces for their identity, even when they are shown side-by-side, they are clearly more accurate if these faces are familiar to them, i.e., if their identity has been encoded in memory [177,178]. There is no denying that: familiarity through experience improves FIR. Yet, human adults also immediately and automatically extract idiosyncratic visual features and configurations of unfamiliar faces (see below), indicating that these faces are not handled by low-level visual processes, as sometimes claimed (e.g., [178,179,180]). That is, there is no need to create this boundary between the recognition of unfamiliar and familiar faces in scientific literature. Indeed, the typical adult human brain recognizes (i.e., provides selective generalizable responses to) the identity of unfamiliar faces much better than children, patients with prosopagnosia, or other animal species [9,10,176], therefore presenting also with a degree of expertise in this function. Moreover, unfamiliar faces are associated with a large inversion effect, another hallmark of face expertise: their recognition accuracy is severely affected by simply presenting the stimuli upside-down ([181,182]; see [183] for review), even if the differences between faces in terms of physical features are equivalent across the upright and upside-down orientations (Figure 16A).

Human adults are also better at recognizing the identity of unfamiliar faces of an experienced as compared to a novel morphology, the so-called “other-race face effect” in the literature [185,186]. In fact, there is no evidence that long-term familiarity with faces changes these latter effects. What changes with familiarity, however, is that we associate semantic knowledge, names, etc., to face identities, and this certainly contributes greatly to our ability to quickly recognize that different pictures of the same familiar face identity belong to the same identity [176,187]. This semantic and verbal knowledge is not available for unfamiliar faces so measuring FIR with such stimuli is invaluable because it allows us to better isolate the contribution of visual features and their configuration to the recognition process.

This is exactly what was performed with the intracerebral recording approach described in the present review [188]. Sixty-nine patients were tested with the paradigm shown in Figure 16A, in which the same face identity (picked randomly in a large pool of unfamiliar faces) appears at a 6 Hz frequency at different sizes, interrupted every five stimuli by a different unfamiliar face identity of the same sex. Stimulation sequences lasted for about one minute, based on the same principle as in the studies reviewed above. Every patient viewed stimulation sequences with upright or inverted stimuli presented in random order (in many cases, no more than two sequences of each orientation). This type of ‘oddball’ paradigm in the context of fast periodic visual stimulation, first reported in EEG by Liu-Shuang and colleagues (2014), is extremely simple and not very original, but valid, highly sensitive, and also highly reliable [189]. For these reasons, this paradigm has been used in more than 25 published studies so far (as reviewed by [52]).

What do we find inside the brain with this paradigm, then? As shown in Figure 16B for a recording contact of one individual brain in the right lateral fusiform gyrus, there is often a clear face individual discrimination (FID in [188], see Figure 17) response (1.2 Hz, 2.4 Hz, etc.) in the SEEG spectrum for upright faces, with a small or even non-significant response for the same stimuli presented upside-down.

Following typical quantification (i.e., the sum of baseline-corrected harmonics), this results in many more recording contacts that are significant for upright than inverted faces, with larger amplitudes for upright faces (Figure 17A).

This is not very surprising: as mentioned above, human adults are much better at recognizing the identity of upright as compared to inverted faces ([181,182]; see [183] for review). Yet, this face inversion effect is not so trivial: although human infants can tell apart upright from inverted faces, the ability to perform FIR tasks better for upright than inverted faces emerges quite late during human development and increases until adulthood ([6,190]). Moreover, it is non-existent or negligible in macaque monkeys or even other non-human primate species [10,140].

Both scalp EEG and fMRI studies have also shown a reduction in identity recognition/discrimination measures (usually through identity repetition paradigms as used here) for inverted as compared to upright faces (EEG: [20,55]; fMRI: [191,192,193]). In the presently described FPVS paradigm, the EEG response amplitude to inverted faces reduces to 47% of the response to upright faces [52]. Critically, this does not mean that 47% of neural FID response is elicited by low-level visual cues (a common misinterpretation). Indeed, inverted faces are structured stimuli that also activate high-level visual regions of the human brain, including face-selective areas of the fusiform gyrus [194] or non-face high-level visual brain regions [195], and these regions may also contribute to the inverted FID response. Nevertheless, these findings show that more than half (i.e., 53%) of the FID EEG response cannot be attributed to objective physical differences between images.

While these effects of face inversion cannot be reliably localized inside the brain with EEG or MEG, in fMRI, they have been tested only in specific functional regions of interest, such as the FFA or OFA. In comparison, in the study of Jacques et al. (2020; [188]) with the large-scale intracerebral approach, this neural face inversion effect was mapped across the VOTC with a direct measure of neural activity. It is important because any difference between upright and inverted faces cannot be attributed to objective physical cues of the stimuli: there is as much difference between pictures of individual faces upright as at inverted orientation. The difference between upright and inverted faces, therefore, reflects our (experienced-based) knowledge of the upright structure of a face. Thus, again, this contrast identifies where our cortical memories of faces are instantiated.

Where are these cortical memories of face identities, then? When we map this inversion effect (i.e., subtracting the responses to inverted faces from upright faces), both in terms of proportion of significant contacts and amplitude, we identify a relatively narrow strip of cortex in the right hemisphere, running from the lateral inferior occipital gyrus (especially for proportion maps) all along the lateral fusiform gyrus, with also a peak in the middle fusiform gyrus in amplitude (Figure 18).

Of note, this high-level FIR neural response extends anteriorly to the middle fusiform gyrus (i.e., anteriorly to the typical FFA location) in the anterior fusiform gyrus but does not reach the most anterior regions of the VATL, close to the temporal pole (Figure 18), presumably due to the lack of association with semantic knowledge for unfamiliar faces. The same localization of neural activity is found in the left hemisphere, albeit with substantially weaker responses. As for the magnitude of the face inversion effect observed in the right IOG and latFG (−53% and −56%, respectively), it is in striking correspondence with scalp EEG results over the right hemisphere in neurotypical adult participants (53%, as mentioned above; [52]). Again, such observations do not only strengthen the validity of the present findings but also of neurophysiological data recordings in temporal lobe epilepsy patients to understand human face recognition and brain function in general, as discussed below.

## 8. Summary, Conclusions, and Perspectives

In the present review, we provided an extensive and structured summary of a 10 year research program aimed at clarifying the neural basis of human face recognition based on the combination of two original approaches: human intracerebral recordings (SEEG) and fast periodic visual stimulation (FPVS).

We focused essentially on cartographies, i.e., operationalizing the function and then mapping it: where is this face recognition function instantiated in the adult human brain? As acknowledged in the review, intracranial recordings of the presentation of face and nonface stimuli have been reported in a number of studies by various research groups since the mid-1990s. However, our approach performed in large samples of patients for 10 years, increasing knowledge step-by-step, has been original at several levels. Most notably, compared to the seminal and highly valuable ECoG studies of Allison, McCarthy, Puce, and their colleagues, in particular in their trilogy of publications in 1999 [71,72,73], (1) we perform recordings intracerebrally, i.e., inside the brain, with SEEG, (2) we rely on FPVS with paradigms that have been well validated in scalp EEG studies of neurotypical individuals, and (3) we objectively identify and quantify periodic neural responses in the frequency-domain. Recording and analyzing the recognition responses is quite straightforward, with only 2–4 stimulation sequences/conditions usually and a minimal number of steps for data transform. Besides objectivity, the frequency-tagging approach provides high SNR and test-retest reliability ([189]; see Figure 14 for evidence in our intracerebral studies; also [37]), which is key to being able to interpret these findings.

Understanding the neural basis of human face recognition is important in itself, but also because the nervous system can be conceptualized as a biological organ of recognition [198], and there are good reasons to consider face recognition as the ultimate recognition function for the human brain [9]. Since there is a substantial degree of human specificity in this function also, this scientific enterprise should help us understand who we are as a species [9].

### 8.1. Spatial Maps of Human Face Recognition

In summary, and specifically, what have we learned so far about the neural basis of human face recognition with these SEEG-FPVS studies?

First, there is an extended spatial distribution of category-selective responses for faces across the human VOTC (Figure 4, Figure 5, Figure 6 and Figure 8). Yet, face-selective responses are not found equally distributed everywhere but follow a certain organization, mainly stretching from the IOG and along the lateral section of the fusiform gyrus (and surrounding sulci), in line with fMRI findings (Figure 4 and Figure 8).

Second, the generic face recognition function, and to a lesser extent face identity recognition, extend to anterior regions of the VOTC, i.e., the VATL, up to the temporal pole (Figure 4, Figure 5, Figure 6, Figure 7, Figure 8, Figure 9 and Figure 18) with no obvious gaps in representation (i.e., a continuous strip of cortex). The gaps between focal regions as identified (in individual brains) in fMRI might be perfectly real, potentially related to local variations in the density of face-selective neural populations. However, we should remain open to the possibility that they may also be partly artifactual due to local drops of magnetic signal (Figure 7). This is particularly the case in the anterior region of the fusiform gyrus, as we emphasized in our earlier review of the present approach (restricted to generic face recognition; [96]) and illustrated in Figure 7. The slightly anterior coordinate of the peak of face-selective activity found in SEEG as compared to fMRI (Figure 6) may be due to the magnetic susceptibility artifact lowering the SNR anteriorly to the FFA (Figure 7). Most importantly, due to this artifact, face-selective activity, as measured in fMRI, may be entirely missing from a large portion of the VOTC in between the FFA and the temporal pole (Figure 7). The issue of magnetic susceptibility artifacts in the (ventral) anterior temporal lobe has been very well acknowledged in the context of general semantic categorization [199] and the neural basis of word reading [95] (Figure 7). However, most fMRI researchers in the scientific community working on face recognition either do not seem to be aware of this limitation or neglect it completely in their data interpretation and models, as if there was a real gap between the FFA in the mid-fusiform gyrus and face-selective regions in the most anterior regions of the VATL, close to the temporal pole (anteriorly to the worst artifact; Figure 7; see, e.g., [100,200]). Importantly, there is no evidence that a coronal slices orientation, as proposed [201], improves SNR in the critical regions where the magnetic susceptibility artifact actually lies (e.g., the anterior fusiform, which was not sampled in the latter study). While distortion-corrected fMRI sequences (e.g., spin-echo data acquisition combined with post-acquisition distortion correction; [202,203,204]; or increased acceleration factor in reducing gradient coil heating; e.g., [205,206]) improve SNR in the VATL, they are rarely if ever used in face localizer experiments. Moreover, these sequences improve SNR concomitantly in other regions, leaving the region of the anterior fusiform gyrus (and neighboring sulci) with the lowest SNR (e.g., Figure 1 in [206]; Figure 1 in [205]). Finally, for a number of researchers, face-selective regions anterior to the FFA/Mid-Fusiform gyrus do not subtend “perceptual (visual) processes” and are therefore either not considered [66] or treated as a higher-order “memory” or “semantic” system, where only representations of familiar faces would be located [200,207]. Although familiar faces, with their rich set of associations, may indeed recruit VATL regions more intensely than unfamiliar faces, our findings of a continuum of intracerebral face-selectivity all along the VOTC up to the temporal pole goes against this gap between “perception” and “memory” functions in human face recognition. In line with a view of memory as a distributed system (rather than different memory systems; [113]), we consider that all of these face-selective populations of neurons form cortical memories of faces, i.e., neuronal populations that have learned through association to respond selectively to certain inputs interpreted as faces. Instead of a gap, in future studies, we expect to disclose a gradient of selectivity of these cortical face memories from physical (i.e., visual) to abstract semantic features all along the VOTC [9].

Third, despite the differences between techniques, our intracerebral findings agree with fMRI studies emphasizing clusters of high amplitude activity [60] and the localization of these clusters, in particular in the right lateral section of the Midfusiform gyrus (“right FFA”). The peak of activity lies in this region, among all VOTC regions (Figure 6 and Figure 8), which had not been identified in previous human intracranial studies of face recognition due to ambiguities in response quantification or a narrow focus on specific ROIs without considering whole cartography. The clustering of face-selective activity is not surprising given that face recognition is so prominent in our species and modern societies and that neurons that fire together wire together [208]. Just how many face-selective clusters there are in the human VOTC is unknown. According to neurofunctional models of the human brain, there should be a fixed number of face-selective clusters, or regions, in all individual human brains, with this number perhaps even conserved across primate evolution [103,128,148,149,209,210]. However, this view of a fixed number of face-selective clusters in every individual brain may rather be due to our theoretical constraints rather than our “innocent eye” in neuroscience [211] (see [212,213]). Another issue is whether these face-selective clusters are functional, i.e., particularly important for the function at stake. Direct intracranial electrical stimulation performed on a subset of the patients whose data are summarized in the present review provides a positive answer to this question (see the recent review of [214] and the brief discussion below).

Fourth, our intracerebral studies confirm that face recognition in the human VOTC is bilateral but with a dominant activity in the right hemisphere (RH). RH dominance of face recognition has been documented since the 1950s [215] with a wide variety of methods (e.g., lesion studies, divided visual field stimulation, neuroimaging, EEG; see the recent review of [90]). Yet, even though this RH dominance of face recognition is not found in non-human primates and is probably specific to the human species, there is no clear evidence that it is due to or related to the left-lateralized language function [90]. In particular, there may be genuine competition between face and visual word representation in neighboring regions of the left VOTC (Figure 15), but evidence that this factor causes the right lateralization of faces is virtually non-existent [90]. Interestingly, our intracerebral recordings studies confirm that RH lateralization is increased for face identity recognition as compared to generic face recognition [216], at least when using unfamiliar faces (i.e., devoid of semantic verbal associations) (Figure 18). This may reflect the fact that (unfamiliar) face recognition is particularly challenging, and thus, considering our relatively large human brain, it requires faster and more efficient local—i.e., intra-hemispheric-processing [217,218].

Fifth, selectivity means specificity: populations of neurons that are category-selective for faces are not category-selective for pictures of houses or visual words (Figure 11, Figure 13 and Figure 14). Even without considering variability in timing or high-frequency broadband activity (see below), the spatial dissociations and intra- vs inter-conditions correlations in amplitudes are so clear that—to use a modern vocabulary in cognitive neuroscience—we could easily use the frequency-tagged category-selective signal to ‘decode’ with very high accuracy what participants are seeing (i.e., faces or houses or words) (e.g., [219]) (Figure 14). Could there nevertheless be ‘information’ about other (visual) categories in (some of) these face-selective populations of neurons identified by our FPVS localizer? Perhaps, and one could probably ‘decode information’ about other object categories using multivariate pattern analyses (MVPA) across face-selective electrode contacts (although this decoding would have to be highly accurate and performed across variable and numerous exemplars of the same category, which is rarely the case in such studies). What would it tell us, though, if, say, there is also information to categorize cars or birds in (some of) the face-selective populations of neurons identified in our FPVS-SEEG localizer? And if, unlike for houses and words, some electrode contacts are active to both faces and birds with a high correlation of amplitude, would it mean that the population of neurons generating this activity is not involved in categorizing (recognizing) stimuli as faces? In fairness, this is not a priority of our research program, which uses faces as a model for understanding how the richest and most complex form of recognition is implemented in the human brain, using widely variable natural stimuli presented with temporal constraints. Moreover, our goal is not to decode the nature of face representation in various regions but to measure the function: how faces are actively (and reliably) distinguished from other object categories, i.e., how they are recognized. Even using only a simple (low) frequency-based analysis, as in all of our studies, we think that the cluster-based right lateralized wide distribution that emerges tells us something about how this function is achieved in the human brain.

Sixth, beyond face-selective activity, we found a large number of recording contacts in regions of the VATL, especially (up to 40%; [60]) with responses to faces only (Figure 8). These face-exclusive responses would be difficult to objectively identify with time-domain analyses only, where all post-stimulus amplitude fluctuations would have to be considered. Thus, even at the level of populations of neurons, exclusive responses to face stimuli can be observed, at least when excluding nonface visual stimuli (i.e., we cannot exclude that these electrode contacts, especially in the VATL, would be sensitive to person-related information in other modalities, such as the voice for instance). Importantly, the increase in the proportion of face-exclusive contacts in more anterior regions of the VOTC does not imply a bottom-up hierarchy of information processing stages insofar as there is no evidence that exclusive responses to faces are built from non-exclusive (selective) responses in posterior regions.

Seventh, in the same vein, our spatial maps provide no evidence that face-selectivity spatially precedes—i.e., is posterior to—identity recognition (compare Figure 6 and Figure 8–Figure 18), as conjectured by a number of MVPA decoding fMRI studies [159,220,221] (but not fMRI-adaptation studies; e.g., [193,222,223,224,225]). In fact, contrary to face-selective activity, sensitivity to differences between identities, for unfamiliar faces at least, does not reach the most anterior VOTC regions (Figure 18). Moreover, the largest proportion of recording contacts that are sensitive to (upright-inverted) face identity in this paradigm also shows face-selective activity. Given that generic face, recognition is much easier and involves a wider population of neurons, this large overlap in neural basis between the two recognition functions is not incompatible with typically preserved generic face recognition in patients with face identity recognition impairments as in prosopagnosia [226].

In summary, despite focusing essentially on cartographies so far, these intracerebral studies reveal a number of findings that makes us progress in our understanding of the neural basis of human face recognition.

### 8.2. General Implications

As discussed above, together with fMRI observations, our findings centered on a specific function, face recognition, provide a model of brain organization for cognition, i.e., not in terms of modules or full distribution but rather in terms of functional peaks of activity, presumably the nodes of an anatomically-defined network. Whether these nodes have a specific subfunction defined by different patterns of connections remains to be determined (see below).

The functional electrophysiological maps derived from the presently reviewed studies performed in epileptic patients are large, if not completely, compatible with fMRI findings in healthy individuals. This is obviously the case for the face-selective peak in the lateral portion of the mid-fusiform gyrus (right ‘FFA’), but also with the right IOG as the second region showing the largest face-selective amplitude (Figure 8) and the spatial distribution of category-selective responses to houses and written words (Figure 11, Figure 13 and Figure 15; see also [170]). Moreover, the magnitude of hemispheric lateralization and stimulation condition effects (i.e., inversion effect) is in line with EEG data recorded in healthy populations (Section 7). Thus, our findings help validate the study of this special neurological population of drug-resistant epileptic patients to understand brain function in neurotypical individuals, at least as far as visual recognition is concerned. Of course, there are serious issues to consider when recording and interpreting such data, such as the presence of lesions, epileptic seizures in the sampled regions, lateralization of epilepsy, effects of medication, fatigue, anxiety, etc. While all these factors certainly combine to lead to absolute amplitude modulation of neural signals, or potential shift of lateralization of function depending on the dominant side of epilepsy [227], there is no evidence of large qualitative changes of function compared to neurotypical individuals. This is in line with neuropsychological evidence that these patients are, on average, slowed down at face recognition tasks, but with no evidence of category-specific effects or qualitative changes (e.g., no reduction in face inversion effect; see [38]).

Finally, in terms of general implications, the compatibility of our findings with fMRI and previous intracranial recording studies strengthens the validity of the FPVS or frequency-tagging approach to measure brain function. In all paradigms described here, stimuli are presented relatively rapidly (e.g., usually with an SOA of 166 ms (for a 6 Hz stimulation rate). This relatively fast rate allows only one fixation per stimulus, reducing contamination by eye movements and potential spurious differences between categories due to different eye movements. While it may be argued that this presentation mode is too fast to capture all of the category-selective (or identity-selective) neural responses, scalp EEG studies show otherwise, indicating that the amplitude of the neural response does not decrease until much faster rates [43,59,169]. Moreover, while periodicity of the stimuli may technically lead to additional anticipatory or predictive processes, scalp EEG studies also show that it is not the case, with face-selective neural responses being identical during a periodic or nonperiodic presentation and rare omissions of periodic face stimuli in trains of objects failing to elicit any deviant ‘oddball’ response [57].

### 8.3. Limitations and Challenges for Future Studies

A number of weaknesses of SEEG studies, as listed in Table 1, can be overcome thanks to recordings performed in large samples and the FPVS approach generating automatic brain responses expressed in a compact frequency-domain representation (sum of corrected harmonic amplitudes) without requiring any challenging tasks for the patients. However, there remain a number of limitations and challenges for future studies.

First, regarding the issue of integrity of function, the epileptic zone in this population often concerns the anterior and medial temporal brain regions. Given that lesions to these regions can cause neuropsychological deficits in semantic associations (e.g., for familiar faces; [228,229,230] for reviews) and learning/recollection of face identity (e.g., [231,232]), future studies of these functions may prove to be more challenging in the population of epileptic patients. More generally, since SEEG recordings are performed in patients with drug-resistant epilepsy and anti-seizure medication, one may question the validity of this approach to provide a model of the normal functional organization of face recognition. In this respect, careful control of the data for each individual patient is mandatory (e.g., exclude recordings in epileptogenic zones or epileptogenic lesions, artifact rejection if necessary, and selection of patients based on minimal neuropsychological criteria). Despite these potential limitations, the evidence accumulated in our studies over the past decade point to a typical functional organization of the VOTC in epileptic patients (highest face-selective responses in the expected regions such as the IOG and latFG, right lateralization of face responses, typical medio-lateral dissociation between faces, houses, and words, typical face inversion effect, etc.). Although a comparison between epileptic patients and normal controls will always be limited by the unavailability of SEEG recordings in non-epileptic subjects for obvious ethical reasons, we expect potential qualitative (e.g., the more frequent shift of hemispheric dominance) and quantitative differences (e.g., reduction in response amplitude). All these issues may be overcome by grouping a large number of patients to blur some of these differences.

Second, the number of implanted SEEG electrodes is minimized, and their localization depends on clinical criteria so that there are relatively few relevant electrode contacts in each individual brain. While this does not prevent the performance of highly informative single case studies when combining electrophysiological recordings and stimulation (e.g., [97,233]; see below), group recording studies as synthetized here can be biased because of under sampling of some anatomical regions. To partially overcome this issue, amplitude maps can be complemented by proportion maps taking into account the number of recording contacts implanted in each region (Figure 15).

Third, while the technique of intracranial recordings is often praised for its high spatial and temporal resolution, in practice, there are limitations to deriving meaningful information at this level. In the group studies presented here, while spatial resolution is technically constrained by the size of the inter-contact distance (3.5 mm spacing along one electrode, center-to-center), it is also limited by the variability in anatomy between individuals and the size of the defined anatomical ROIs (which must be sufficiently large to encompass a number of contacts providing a reliable response). Moreover, as mentioned above, electrical fields spread in the brain tissue and can be measured at a distance from the actual neural source of the field. In this respect, a potentially important issue concerns the reference channel used to measure these face-selective potentials at each recording contact. In all studies reviewed here, an electrode contact in the distant white matter (WM) or (for a majority of contacts) on the scalp (FCZ) is used as a reference. Most intracranial studies, in ECoG in particular, also used a physical electrode as reference (scalp, subdermal, cranial, or intracranial electrode), and the intracranial data are typically analyzed with a common average reference. While the use of a common average reference is difficult to justify given the spatial inhomogeneity of the sampling, especially in SEEG, there may be advantages in using a local bipolar montage to increase spatial resolution by limiting the effects of electrical field spatial diffusion. However, by and large, changing the reference montage to a common average reference or a bipolar montage has little effect on the face-selective spatial maps reported here and the relative contributions of the different VOTC regions to face recognition (Figure 19; unpublished data).

Fourth, as indicated earlier, iEEG provides rich and complex data sets, first because of the high spatial and temporal resolution (see below for a discussion about temporal aspects) but also because of the multiplicity of the recorded neural signals. At a macroscopic level of organization, there are two prominent neurophysiological signals: phase-locked signals in the lower range of the frequency spectrum (<~30 Hz, corresponding to the classical event-related potentials and often called local field potentials in SEEG), which were the focus of the studies previously described in this article, and non-phase-locked signals in the higher frequency range (>30 Hz also known as “gamma range” or high-frequency broadband activity, called HF here; see [82]). A major challenge for the human SEEG approach is to determine which of these signals are most meaningful to map and understand human brain function. Nowadays, the neuroscientific community has largely shifted its interest to HFB signals, which are thought to reflect more local neural activity [234,235,236] and population-level neuronal firing [236,237,238], correlates with blood oxygen level-dependent (BOLD) activity as recorded with fMRI [88,235,239,240] and are more straightforward to characterize the response timing (i.e., avoiding the issue of varying polarity and morphology of ERP responses, see below).

However, it is important to note that the FPVS approach is not restricted to low-frequency signals and can also be applied to objectively identify and quantify HFB responses (Figure 20A). In this context, we recently provided a large-scale, in-depth investigation of the spatial and functional relationship between these two signals based on the face-selective responses recorded with SEEG from 121 subjects [37]. Overall, we reported highly similar spatial patterns of face-selective low-frequency and HF neural activity (Figure 20B), including relative local spatial extent, an overall reduced face-selective HF amplitude in the vATL, strongly shared functional properties (corresponding amplitudes, see Figure 20C, face-selectivity) and similar timing characteristics. While these findings point to largely similar properties between the two types of neural signals, our results point to a major quantitative difference, with low-frequency category-selective activity having a higher signal-to-noise ratio, which allows more extensive exploration of the vATL. This suggests that iEEG studies, in general, should not only focus on HFB signals but must also consider low-frequency signals, especially when recording in or above the vATL.

Fifth, such as ECoG, SEEG recordings provide both high spatial (in the order of millimeters) and temporal (in the order of milliseconds) resolution recordings of neural events, allowing, in principle, to track the spatio-temporal dynamics of human face recognition (e.g., [78,79]). While the strength of FPVS is its objective and high signal-to-noise ratio measurements when using, frequency-domain analyses, the FPVS approach can also be used to take advantage of the high temporal resolution of the SEEG neurophysiological signal, as long as the temporal distance between significant events (here faces) is long enough to avoid overlapping neural (differential) activities [43]. For the face ‘localizer’ paradigm shown in Figure 2, this can be achieved by dividing the (~60–70 s) recording sequences into smaller epochs around the time at which faces are presented, i.e., every 0.833 s (i.e., 1.2 Hz periodicity) and averaging the resulting epochs for each recording contact ([43,54,63] for scalp EEG). This was conducted in our recent study on 121 individual brains, comparing face-selective neural signals in low and high frequencies [37], segmenting either the raw SEEG signal (for low frequencies, LF) or the HFB amplitude envelope (for high frequencies, HF, Figure 20A). To isolate the face-selective components of the response, the signal around the frequency of the base stimulation (i.e., 6 Hz and harmonics) was filtered-out (using narrow notch filters) from the LF signal and the HF amplitude envelope. In addition, to get rid of the large variability in the morphology and polarity of the LF responses (i.e., ERP components) across contacts, regions, or patients/individuals (e.g., [78,79]), the LF signal was converted to phase-free signal amplitude modulation using a Hilbert transform. This analysis revealed the striking similarity in response onset time between LF and HF face-selective signals in the latFG, although LF signals were more sustained in time (Figure 21). In future work, this approach will be used across various brain regions to define the relative onset/offset time and time-course characteristics of category-selective responses during FPVS.

Sixth, while the FPVS approach could potentially be used in combination with SEEG recordings to map other face recognition functions (e.g., emotional facial expression, gender recognition; see [241,242]), visual recognition in general or recognition in other sensory modalities, the present review has not addressed the issue of the criticality of the mapped brain regions for recognition functions. This issue can be addressed by the complementary application of small electrical currents to electrode contacts—Direct Electrical Stimulation (DES)—an approach typically used for clinical purposes (i.e., determining the critical functionality of brain regions candidate for surgery). Early ECoG studies noted facial hallucinations and transient disruption of famous face naming in several individuals following DES to various VOTC sites [70,73]. However, it is fair to say that behavioral FIR during DES with ECoG (see also [91,243,244]) has not been thoroughly studied, objectively quantified, and related to neural measures of FIR (but see [245,246] for quantified behavioral effects of DES to the FFA on generic face recognition). In contrast, over the last decade, our group has provided detailed behavioral reports of a few patients who, upon intracerebral DES of face-selective regions of the (right) VOTC, were suddenly unable to recognize the identity of faces (Figure 22). One such case, in fact, the first reported case of transient prosopagnosia, was found following DES to the right IOG/OFA: patient KV was suddenly and transiently unable to recognize the identity of famous faces (reporting various face-specific phenomenological experiences) ([247]) or to discriminate different identities of unfamiliar faces [248]. Another patient, MD, when stimulated inside the right FFA, showed transient face identity palinopsia, perceiving facial features of an unidentified identity incorporated in the currently presented face identity (a real person’s face or a picture) [233]. Last but not least, focal DES to face-selective sites of the right anterior fusiform gyrus appear to elicit transient prosopagnosia slightly more often, with two cases having been reported in detail [97,249]. The second case, patient ND, is particularly interesting since he excelled at FIR outside of stimulation and was tested during DES to the right anterior fusiform gyrus with a wide variety of behavioral tasks of FIR, including nonverbal celebrity pointing tasks and matching tasks on both familiar and unfamiliar faces [97]. In truth, such observations remain extremely rare—as are real cases of prosopagnosia following brain damage (see [176,250])—and a full account is beyond the scope of the present review (see the reviews of [214,251]). However, besides the fact that these effects always occur during stimulation of the right but not the left VOTC, it is worth pointing out in the context of the present review that they are also tightly linked to face-selectivity and sensitivity to face identity in all cases, as particularly evidenced when using the amplitude of the frequency-tagged neural activity [97,233,248] (Figure 22). Importantly, while these rare effects occur only to focal stimulation (i.e., usually only to a single pair or two adjacent pairs of electrode contacts), this does not mean that the effect is only focal since the different face-selective VOTC regions are thought to be interconnected anatomically and functionally [102]. These stimulated local sites could be considered key nodes of the face recognition network, allowing the stimulation effect to spread to other connected regions [245], perhaps a necessary condition to elicit behavioral impairments [251]. DES during concurrent FPVS stimulation in future studies could be applied to demonstrate effective connectivity between these regions, in particular of these key nodes.

Finally, although SEEG provides recordings with a very high spatial resolution compared to other methods, we made it clear in the present review that the electrode contacts (usually cylinders of 0.8 mm in diameter and 2 mm in length) nevertheless capture the pooled activity of hundreds of thousands of neurons or more. However, thanks to hybrid macro-micro electrodes (typically SEEG electrodes with microwires outgoing from the end of the electrode, e.g., see [252]; Figure 23A), SEEG provides the opportunity to also record multi-unit (MU) and single unit (SU) spiking activity. This rare approach in humans [253] has led to highly original observations - such as the so-called ‘concept cells’ firing to multimodal inputs related to people, places, or events in the human medial temporal lobe [254,255]. However, it is particularly challenging to perform in the VOTC because there is little cortical depth for the tip of the electrode, where microwires are located, to end up in the grey matter (Figure 23A,B).

In the past few years, multi and single-neuron activity during face recognition has nevertheless been recorded from the human occipito-temporal cortex of single individuals [256,257,258] or even several individuals (N = 8 in [259]). Given the above-mentioned limitations of the monkey model to understand the neural basis of human face recognition, these action potential recordings can potentially be extremely informative. Given the diversity of responses – varying in terms of onset, duration, excitatory or inhibitory observed in these first recordings in or near face-selective areas [259], here again, we think that a non-conventional FPVS approach will be particularly useful (e.g., Figure 23C,D,E), providing highly sensitive and objective responses (i.e., no subjective definition of time-window to define spike rates, baseline activity computed on neighboring frequency bins of the tagged frequency) to understand how single neurons fire selectively to faces in general and to specific individuals in these human VOTC regions.

**Figure 23 brainsci-13-00354-f023:**
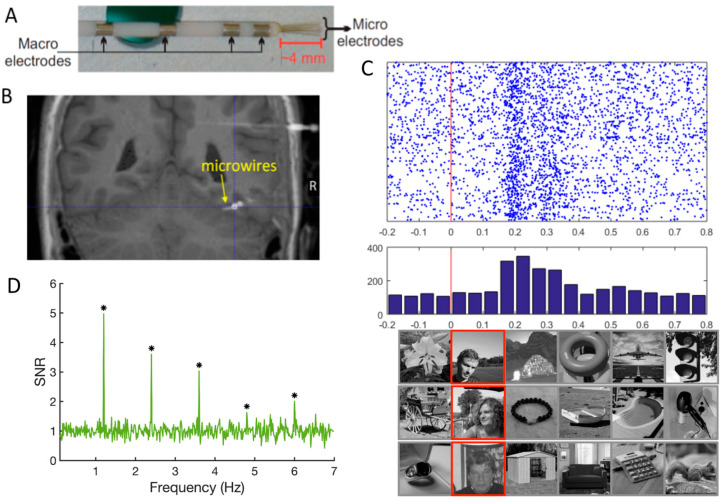
(**A**,**B**) Electrode contacts to record neuronal activity in face-selective vOTC regions, with microelectrodes at the tip of the electrode. (**C**) Selective increase in spike rates in the MidFusiform Gyrus synchronized to the onset (red vertical line) of various face stimuli (see examples in red boxes in lower panel) in a 6 Hz train of objects (1 patient, two trials of 60 s, unpublished data). Each row depicts the raster (each blue dot is a spike) of a 1 s epoch (−0.2 s to 0.8 s) in one of the continuous 60 s periodic stimulation, showing here an excitatory response emerging at around 150 ms post-face onset. (**D**) Frequency-domain response (green) based on the whole 60 s stimulation window shows readily quantifiable face-selective neuronal spiking activity at 1.2 Hz (with harmonics, 2.4 Hz, etc.) and a small 6 Hz response common to all stimuli. Asterisks depicts significant responses at relevant harmonics. Microelectrode-recorded signals are processed by band-pass (300–6000 Hz) non-causal filtering, threshold-based spike detection, and artifact removal. For MU activity, remaining transients are convolved by a 20 ms gaussian window, and the FFT of the resulting signal is computed in order to evaluate periodic variations of the firing rate of the neuronal population. For SU activity, spike events are clustered with sorting algorithms [260]. Single unit spike trains are convolved with a rectangular/gaussian kernel, and the FFT of this signal is then computed.

## Figures and Tables

**Figure 1 brainsci-13-00354-f001:**
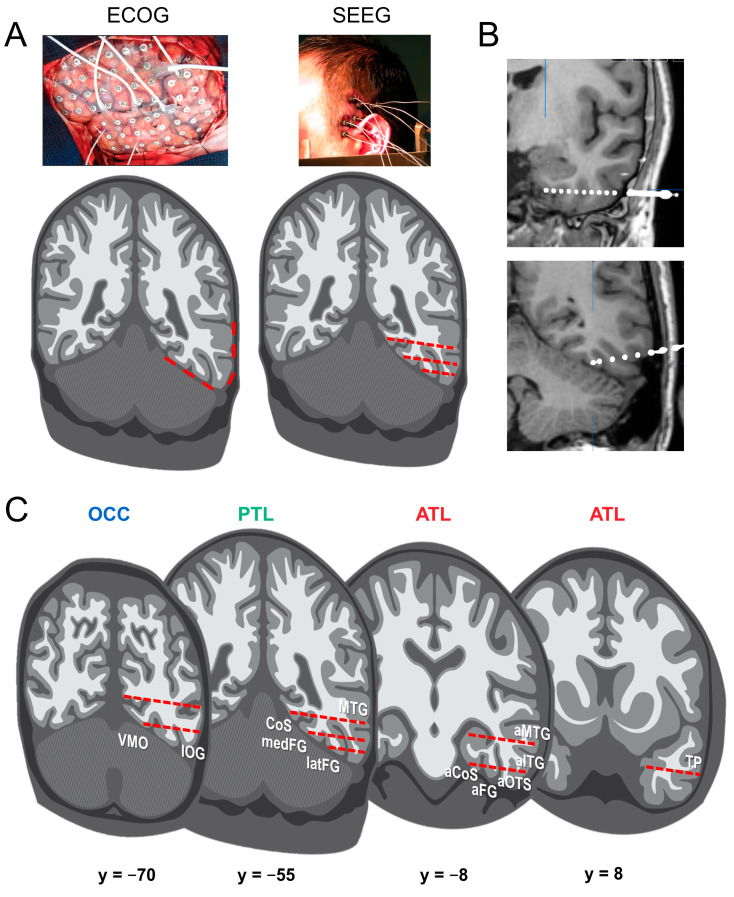
(**A**) Intracranial recordings in human epileptic patients are performed either with subdural grids of electrodes (Electro-CorticoGraphy, ECoG, on the left) or electrodes inserted inside the cortex and subcortical structures (Stereo-ElectroEncephaloGraphy, SEEG, on the right). In ECoG, part of the skull is removed to apply electrodes onto the cortical surface (here, grids of electrodes; see lower row for example electrode locations, in red, on a schematic coronal slice). In SEEG, small holes are drilled in the skull to implant thin depth electrodes. Lower row: Recording contacts from typical electrode trajectories are represented in red. The two techniques differ in terms of their advantages, both in terms of clinical investigations and associated research purposes. The present review focuses on SEEG, which is growing in usage across the world for clinical reasons (less invasive since there is no craniotomy, reduced incidence of infections, and hemorrhages; [31,34,35]. (**B**) Examples of SEEG electrodes in two patients as they appear when co-registering the pre-operative MRI with the post-operative CT-scan (white ‘dots’ on the coronal slices; top: electrode in the ATL, bottom: electrode in PTL targeting the latFG). Each electrode typically contains 8–15 contiguous individual recordings sites (or contacts). (**C**) Typical locations of electrodes implanted (across patients) in the ventral occipito-temporal cortex, as reviewed here (OCC: occipital; PTL: posterior temporal lobe; ATL: anterior temporal lobe. See below for abbreviations of specific cortical structures).

**Figure 2 brainsci-13-00354-f002:**
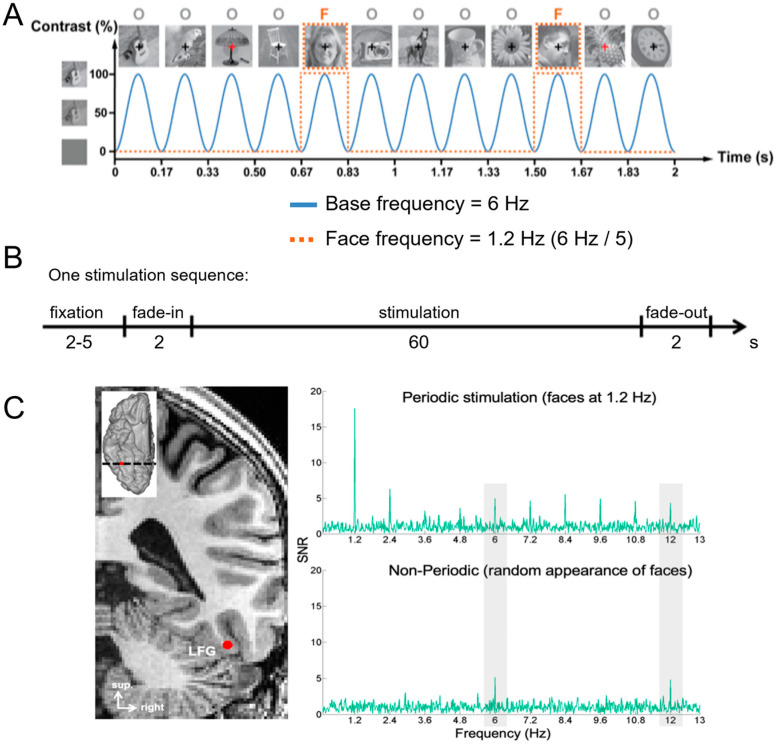
(**A**) Experimental FPVS paradigm as used in our laboratory since December 2012 to record face-selective intracerebral activity in human subjects. Images of objects are presented by sinusoidal contrast modulation at a rate of six stimuli per second (6 Hz, depicted as a blue sine wave), with a different face image presented every five stimuli (i.e., appearing at the frequency of 6 Hz/5 = 1.2 Hz; show in orange boxes; see [54] for paradigm details). O: objects, F: faces. Typically, the neural responses are recorded implicitly, in the sense that participants are not asked to recognize faces (usually they are instructed to fixate a cross appearing over the images and detect random color changes of the cross (e.g., black to red) during a ~60 s stimulation sequence as in (**B**)). (**C**) Objective and high SNR intracerebral responses in the VOTC (right lateral Fusiform gyrus, latFG) of a single individual brain tested with the paradigm depicted above. An SNR of 2 means a 100% increase in signal as compared to noise. The location of the recording contact (indicated by a red dot) is shown using a postoperative CT co-registered to a preoperative MRI. Above, significant face-selective responses exactly at the face-selective frequency (1.2 Hz) and harmonics (up to 10.8 Hz) are observed. If the same (number of) faces appear randomly in the 6 Hz stimulation sequence, Fourier analysis does not identify these face-selective responses at specific frequencies. SNR spectra are obtained by dividing the amplitude at each point of the spectrum by the average amplitude of the neighboring frequency bins [55].

**Figure 3 brainsci-13-00354-f003:**
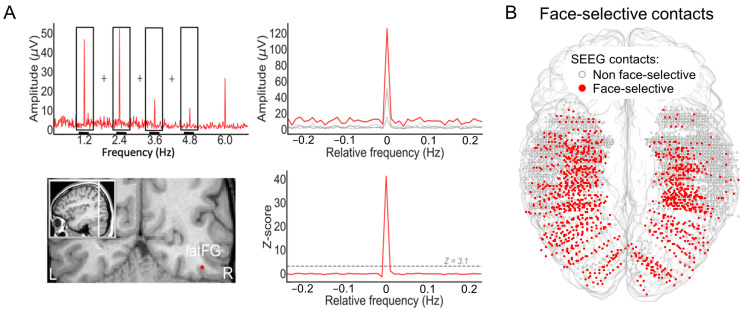
(**A**) Procedure used to define significant face-selective recording contacts inside the brain. Top-left: Intracerebral response measured at a single recording contact (see location on coronal slice – bottom-left) in the frequency domain is show in red. Response significance in defined for each recording contacts typically based on the amplitudes at their first four harmonics (up to 4.8 Hz here). Quantification takes into account more harmonics [37,60] even though the bulk of the signal is found at the first harmonics [43,52]. The frequency spectrum is segmented into four segments containing the amplitudes at relevant harmonics and neighboring frequency bins (top-left, depicted as black rectangles). The spectrum segments (gray lines in top-right panel), containing the signal and the surrounding noise are summed (red line in top-right panel), and a Z-score is then derived at each frequency bin based on the iEEG noise estimated from neighboring bins [62]. (**B**) Significant contacts (e.g., Z-score > 3.1) across individual brains (here, N = 84) are displayed on a template brain (Colin27 brain in the Talairach space).

**Figure 4 brainsci-13-00354-f004:**
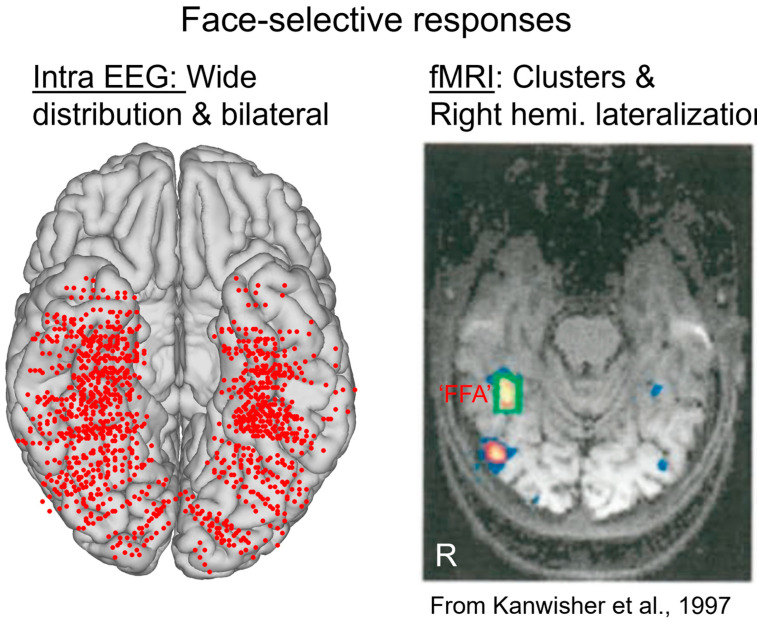
Illustration of the contrast between—on the left—iEEG findings of face-selective activity in the VOTC as described here, with a wide bilateral distribution (N = 84) and—on the right—the local face-selective regions, or clusters, found in fMRI (here reproduced with permission from the seminal study of Kanwisher et al., 1997 [67] usually with right lateralization.

**Figure 5 brainsci-13-00354-f005:**
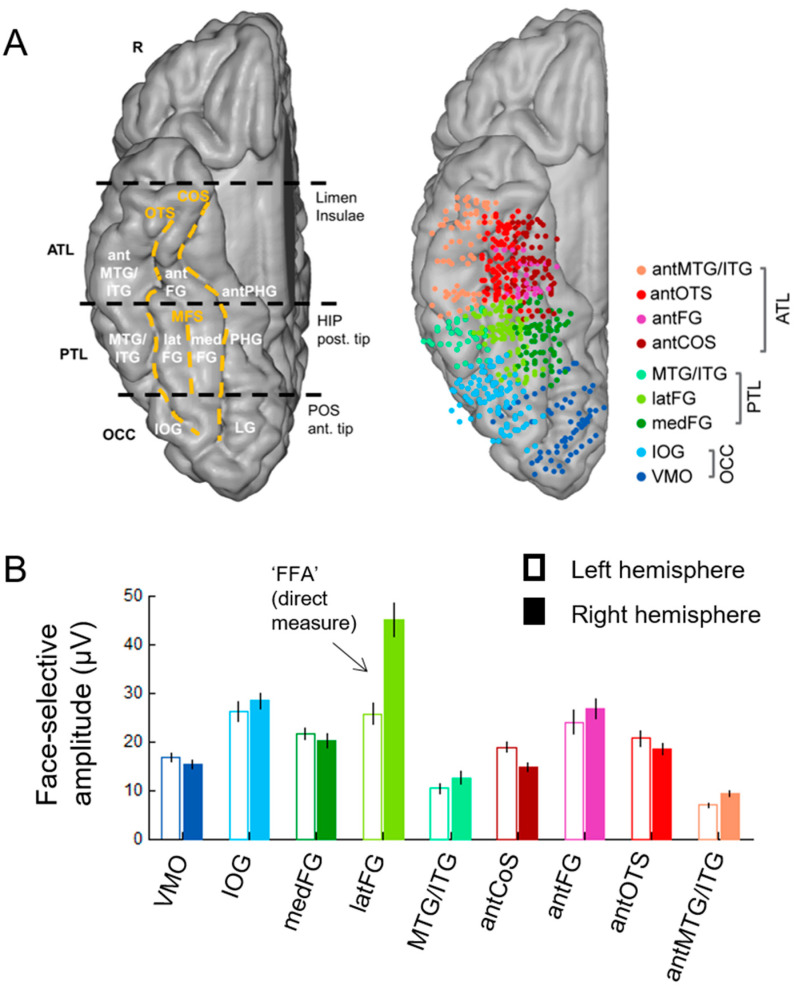
Individual anatomical localization of SEEG electrodes. (**A**) Left panel an example of a VOTC parcellation as used in Jonas et al. (2016) [60]. Major VOTC sulci (show as yellow dashed curves) served as mediolateral landmarks (CoS and OTS), and coronal reference planes containing given landmarks served as posteroanterior landmarks (shown as straight black dashed lines). A coronal plane, including the anterior tip of the parieto-occipital sulcus, served as the border of the occipital and temporal lobes. A coronal plane, including the posterior tip of the hippocampus, served as the border between PTL and ATL. Right panel, face-selective contacts (across 84 subjects) in the Talairach space colored according to their anatomical label in the individual anatomy. (**B**) Average amplitude of the face-selective activity in the different anatomical regions of interest as defined in each individual brain (N = 84). Note the much larger amplitude found in the lateral portion of the fusiform gyrus (latFG) in the right hemisphere. Face-selective contacts were grouped by anatomical ROIs across all participants, and the face-selective amplitude was averaged across contacts to obtain the mean response amplitude for each region separately for the left and right hemispheres.

**Figure 6 brainsci-13-00354-f006:**
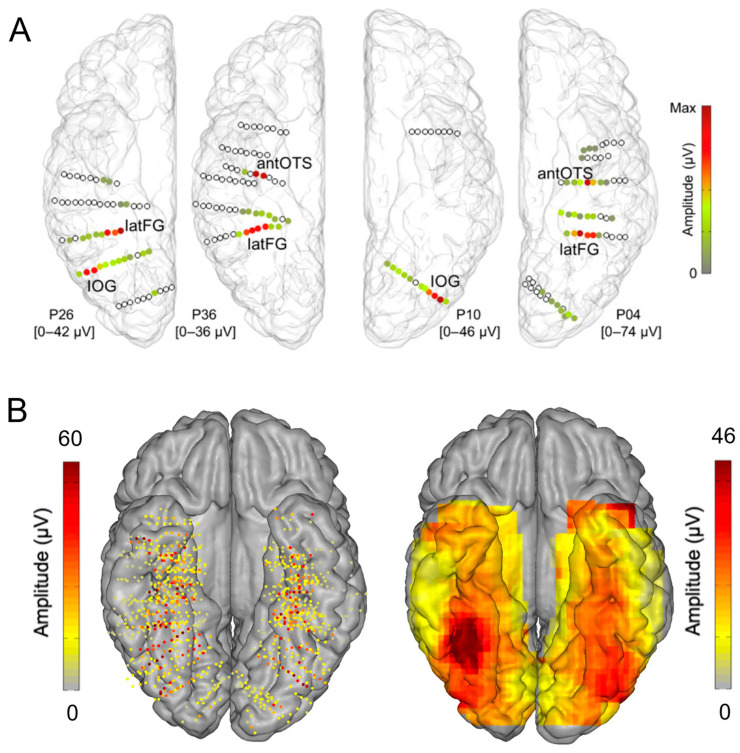
(**A**) Quantification of face-selectivity in individual brains Individual level. Contacts were color-coded according to their relative face-selective amplitude in four participants. (**B**) Quantification of face-selectivity at the group level in a common space. Significant contacts across 84 subjects were color-coded according to their relative face-selective amplitude and displayed in the Talairach space (left panel). Amplitude across contacts can also be smoothed and projected on the cortical surface by plotting the mean face-selective amplitude within 15 mm × 15 mm voxels (right panel). The peak of activity is found at Talairach coordinate x = 38, y = −47, z = −15, which is only slightly anterior to the average coordinate reported in the seminal study of Kanwisher et al. (1997) [67] (x = 40, y = −55, z = −10), and in a more recent fMRI study with the same stimuli and frequency-tagging approach (Gao et al., 2018 [69]: x = 42, y = −54, z = −14).

**Figure 8 brainsci-13-00354-f008:**
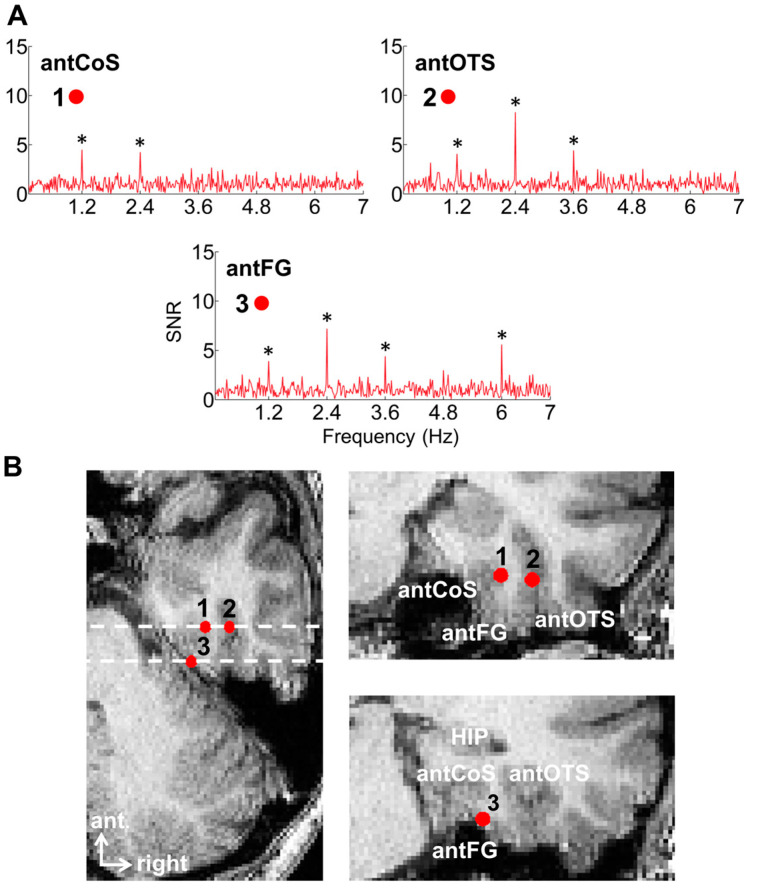
Example of face-selective responses in three distinct anatomical regions of the ventral ATL (from Jonas et al., 2016 [60]). (**A**) Face-selective responses were recorded from the right antCoS, antOTS, and antFG in a single brain. Note that in the examples shown here for the antCoS and antOTS, no general visual responses were recorded at 6 Hz and harmonics (“face-exclusive” responses). * Statistically significant responses (Z > 3.1, *p* < 0.001). (**B**) Anatomical locations of corresponding recording contacts on MRI slices. Contacts are shown as red dots on axial (**left** panel) and coronal (**right** panel) slices. White dashed lines over the axial view (**left** panel) show the location of the two coronal slices (**right** panel). Electrode contacts 1, 2, and 3 are, respectively, located in the antCoS, antOTS, and antFG. The antFG is located between the antCoS and antOTS at a level where the hippocampus (HIP) is visible on a coronal slice.

**Figure 9 brainsci-13-00354-f009:**
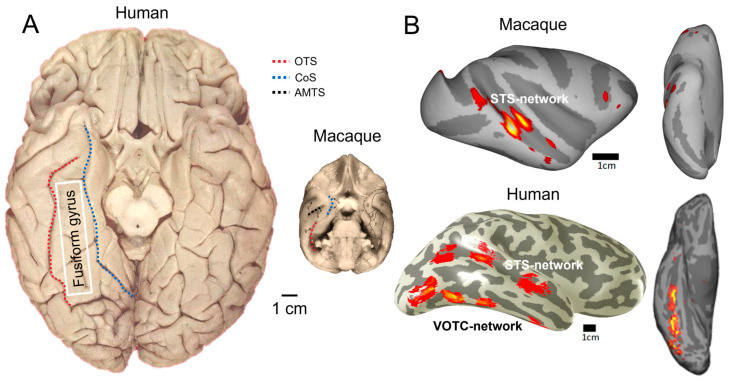
Human specificity of neural circuits for face identity recognition. (**A**) Ventral view of a human and a macaque brain at relative sizes. The human brain contains an estimated number of 86 billion neurons, about 13.5 times more than a macaque brain (6.3 billion) [143]) In humans, the key neural structures for face identity recognition run from the lateral section of the inferior occipital gyrus to the posterior, middle lateral and anterior fusiform gyrus (FG) up to the temporal pole. In comparison, the posterior ventral occipito-temporal cortex of the macaque brain is limited to one sulcus, with little gyrification and no fusiform gyrus [147]. (**B**) Inflated segmented brains showing typical locations of posterior face-selective regions (colored spots) in macaques and humans (adapted from [103]; with permission under a PMC Creative Commons License). In macaques, these regions are essentially found in the STS, whereas humans have face-selective regions in both the STS and the VOTC.

**Figure 10 brainsci-13-00354-f010:**
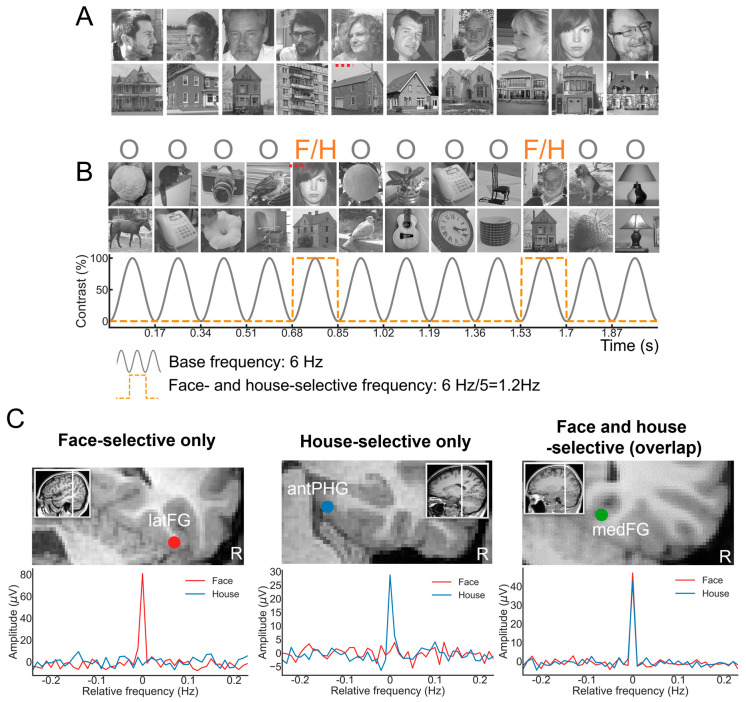
(**A**) Example of natural images of faces and houses used in the experiment of Hagen et al., 2020 [152] (actual face images not shown for copyright reasons). (**B**) Images of living or non-living objects (O) were presented by sinusoidal contrast modulation at a rate of six stimuli per second (6 Hz), with different images of either faces (F) or houses (H) presented in separate sequences at 1.2 Hz. (**C**) Examples of baseline corrected FFT spectra (in different participants) show a face-selective contact with no response to houses at all, a house-selective contact with no response to faces at all, and a contact with face and house-selective responses (overlap contact).

**Figure 11 brainsci-13-00354-f011:**
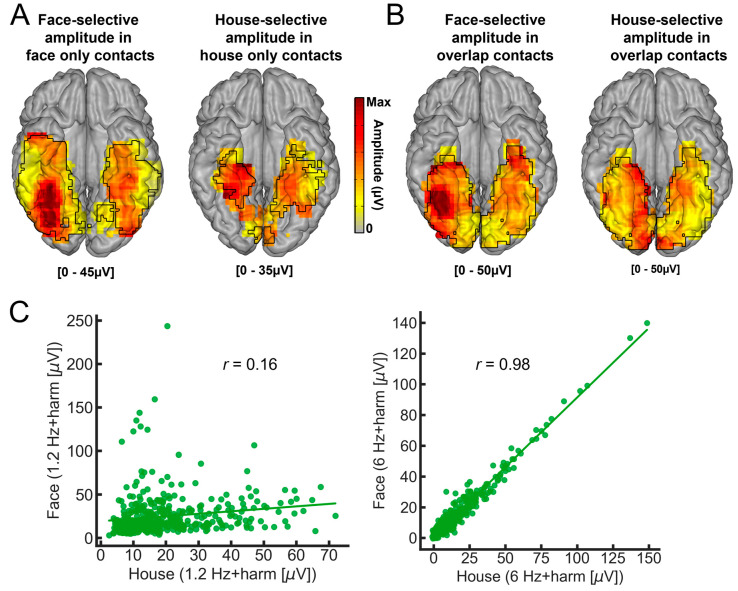
(**A**) Color-coded smoothed spatial maps of the local amplitude measured at selective contacts for only one category, face-selective or house-selective contacts (see examples in Figure 10C, displayed on the ventral cortical surface (from [152]). Note the clear spatial dissociation between the localization of face- and house-selective contacts, with house-selective responses located mainly in medial occipito-temporal regions. (**B**) Spatial maps of the local amplitude -of selective-contacts for both categories (overlap contacts), separately for the face- and house-selective responses. The spatial dissociation is also found at this level, with lateral (faces- and medial (houses) relative distributions of amplitudes. (**C**) For these overlap contacts, despite a ceiling correlation for the base rate response, there was virtually no correlation in amplitude between the response recorded for faces and houses, suggesting that the category-selective responses are generated by different populations of neurons.

**Figure 12 brainsci-13-00354-f012:**
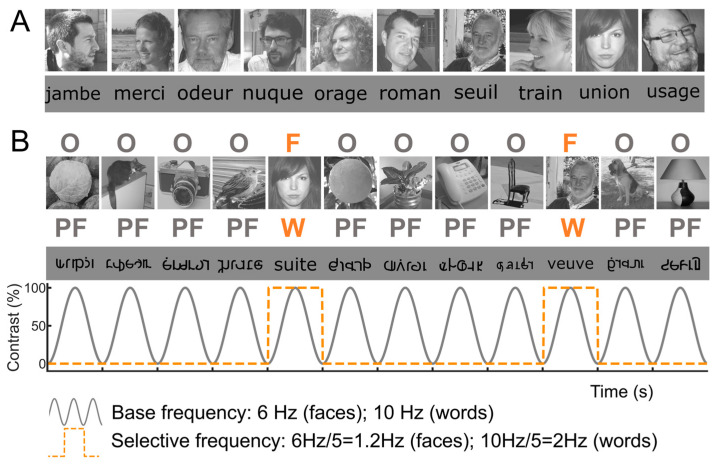
Comparing sequences of face (F) stimulation among objects (O) to written words (W) among pseudofonts (PF) [168]. (**A**) Example of images of faces and words (French). (**B**) Images of faces or words objects were inserted in sequences of objects and pseudofonts, respectively (1 out of 5 images).

**Figure 13 brainsci-13-00354-f013:**
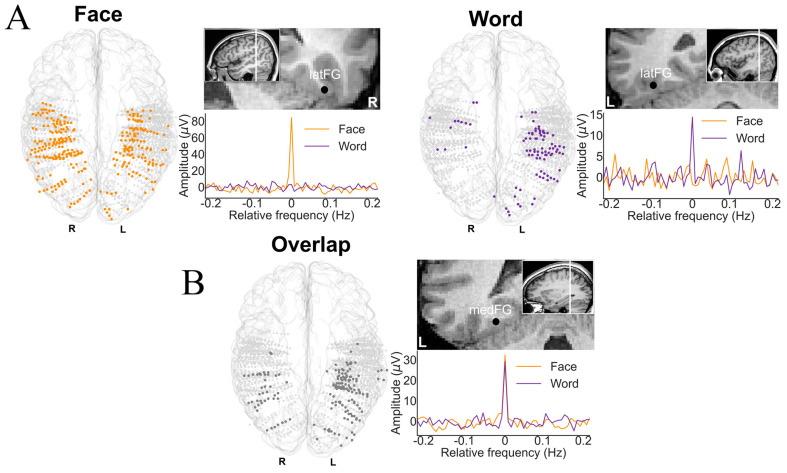
From Hagen et al., 2021 [168]. (**A**) respective localization of significant face-selective contacts with no selective response to words, and vice-versa (with example contacts illustrated). Note the larger number of significant contacts for faces, with an RH advantage, and the large LH dominance for word-selective contacts (as reported in [170]). (**B**) Distribution of contacts showing significant responses to both categories, mainly in the left VOTC.

**Figure 14 brainsci-13-00354-f014:**
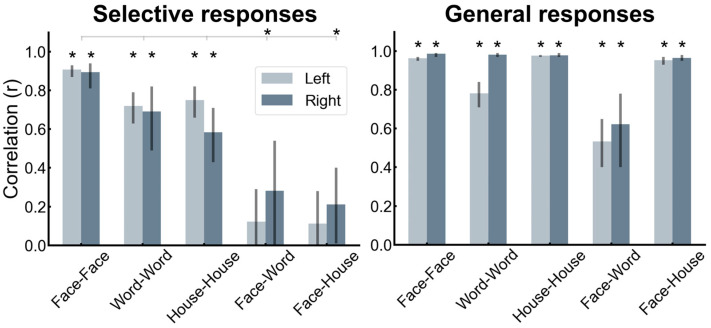
**Left** Panel: correlation values between category-selective amplitudes measured at overlapping electrode contacts either within-categories (first 3 bars; asterisk indicates correlation significantly > 0) or between faces and words (4th bar) showing little to no correlation (no higher than faces and houses, 5th bar) for these two categories. These low correlation values are statistically lower than within-categories correlation values). The high within-category correlation also illustrates the high test-retest reliability of these measures (especially for face-selective activity across two stimulation sequences only). In all conditions, the general visual responses (**right** panel) are also highly correlated (except for faces and words, with very different stimuli appearing at the base rate: objects and pseudofonts, respectively) [168].

**Figure 15 brainsci-13-00354-f015:**
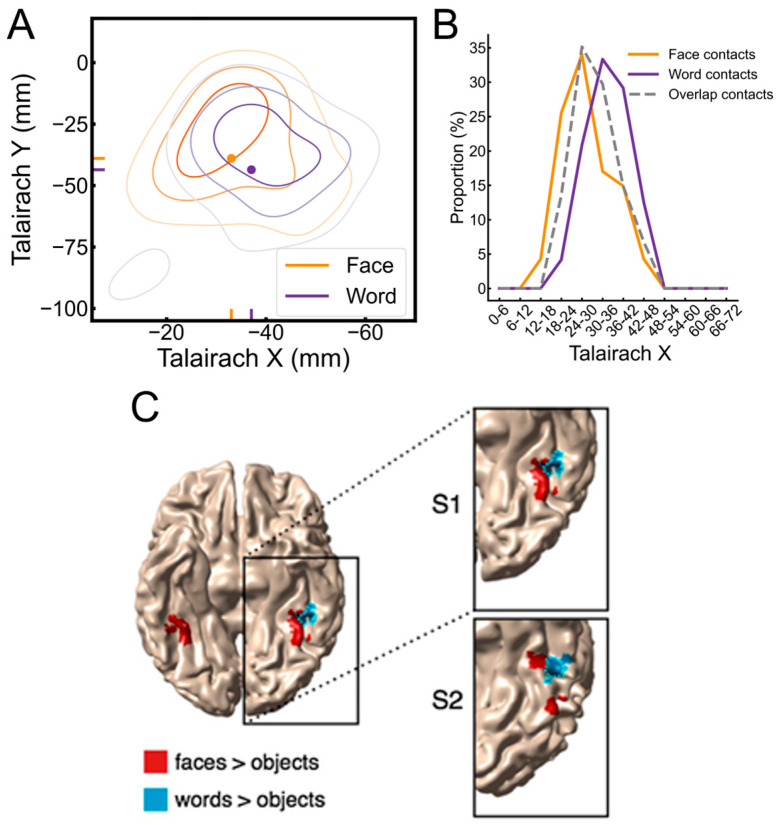
Spatial distribution of word- and face-selective responses in the left hemisphere [168]. (**A**) Left: contour plot showing the distribution of face- and word-selective SEEG contacts in the left hemisphere along the medio-lateral (Talairach X) and posterio-anterior axes (Talairach Y). Each distribution is normalized relative to itself, and darker contour colors indicate a larger density of contacts. The central mass for each distribution is plotted as a solid dot within the contour plot and as a vertical/horizontal line on the *x*- and *y*-axis. (**B**) X coordinates (medio-lateral) of the word- and face-selective contacts in core regions for neural processes supporting visual word and face recognition (IOG, FG + sulci). Word-selective responses in the fusiform gyrus are slightly more lateral on average, as also found in fMRI in neurotypical individuals (illustrated in (**C**), from [171]).

**Figure 16 brainsci-13-00354-f016:**
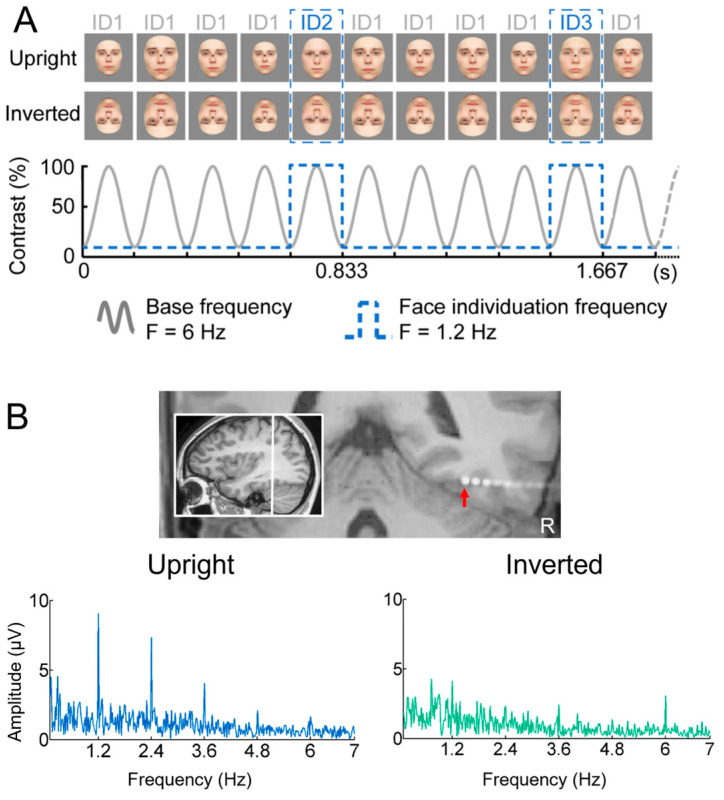
(**A**) The basic FPVS paradigm to measure (unfamiliar) face identity recognition (as published originally by [184]). The same face identity (ID1) is repeated at 6 Hz (across substantial changes of size or other low-level characteristics), interrupted every five stimuli (i.e., at 1.2 Hz) by different face identities (ID2, ID3, …). Inversion—preserving all physical feature differences between the stimuli—generally acts as an additional control to isolate the contribution of our experienced-based structural memory of faces to the recognition function. (**B**) SEEG frequency-domain responses recorded at an individual intracerebral recording contact in the upright (left) and inverted (right) conditions depicted above, with much larger responses to upright faces. The anatomical location of the contact (in the right latFG) is shown in a coronal MRI slice (indicated by a red arrow).

**Figure 17 brainsci-13-00354-f017:**
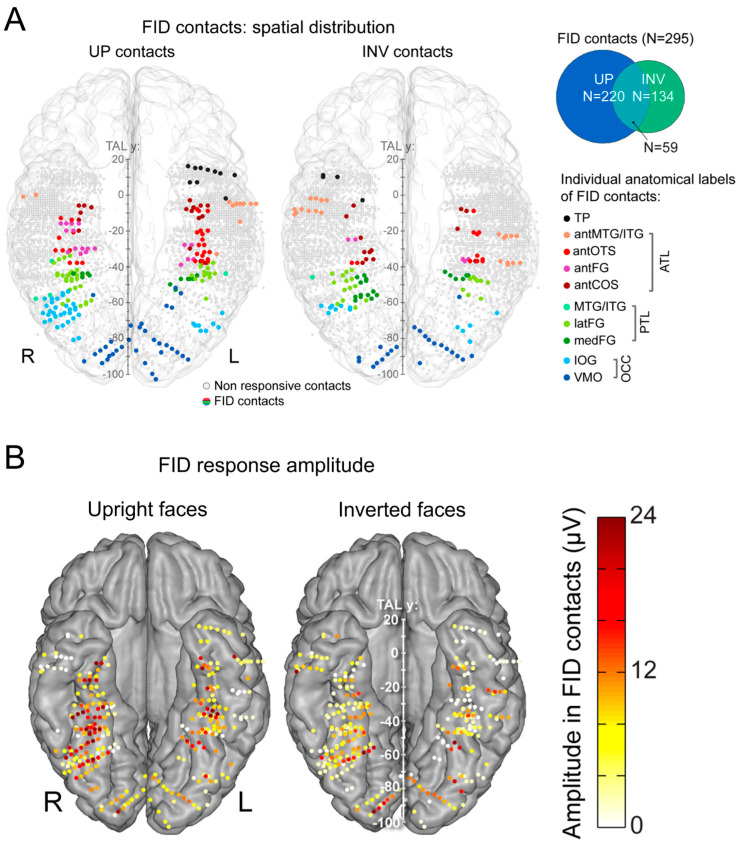
(**A**) Maps of all 3825 VOTC recording contacts across the 69 individual brains tested in the study of Jacques et al. (2020) [188]. The maps are displayed in the Talairach space using a transparent reconstructed cortical surface of the Colin27 brain (ventral view). Each circle represents a single recording contact. Color-filled circles correspond to face identity discrimination (FID) contacts, either defined as UP contacts (left map, N = 220, see Venn diagram inset on the right) or INV contacts (right map, N = 134). UP and INV contacts are color-coded according to their anatomical location in the original individual anatomy (see the legend on the right). White-filled circles correspond to contacts on which no significant FID responses were recorded. For visualization purposes, individual contacts are displayed larger than their actual size (2 mm in length). Values along the Y axis of the Talairach coordinate system (antero-posterior) are shown near the interhemispheric fissure. (**B**) Significant FID contacts color-coded according to their respective amplitude for both upright and inverted faces.

**Figure 18 brainsci-13-00354-f018:**
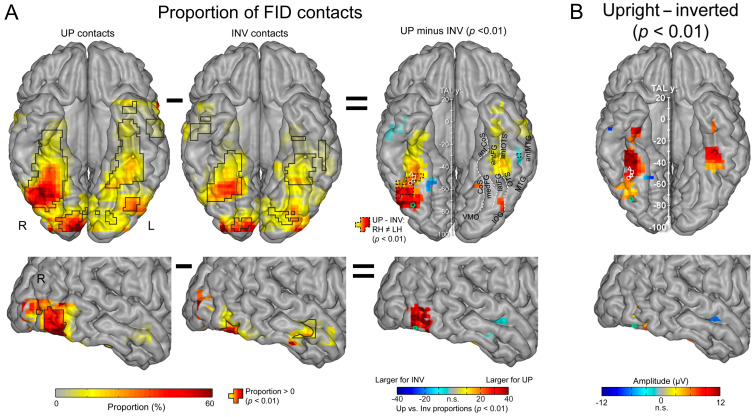
(**A**) Ventral and right lateral maps of the local proportion of significant UP contacts (left) and INV contacts (middle) relative to a number of recorded contacts, and statistical comparison between the local proportions of UP and the local proportion of INV contacts across VOTC (right). For left and middle maps, solid black contours outline proportions significantly above zero at *p* < 0.01. For the ventral map on the right, the dashed black contour lines show the location of the proportion difference between UP and INV contacts that are significantly (*p* < 0.01) larger in one hemisphere. This indicates that the higher proportion of UP compared to INV contacts in the IOG and latFG is significantly larger in the right hemisphere. The coordinates of the published peak location of face-selective regions are shown in white for fMRI-defined FFA (star: [196], circle: [69], square: [65]), in green for fMRI-defined OFA (square: [65], diamond: [197]) and the mean coordinates for latFG face-selective responses in a recent SEEG study are displayed with a white triangle ([60]). (**B**) Ventral and right lateral maps showing smoothed response amplitude over FID contacts displayed over the VOTC cortical surface for significant amplitude difference (*p* < 0.01) between upright and inverted face conditions.

**Figure 19 brainsci-13-00354-f019:**
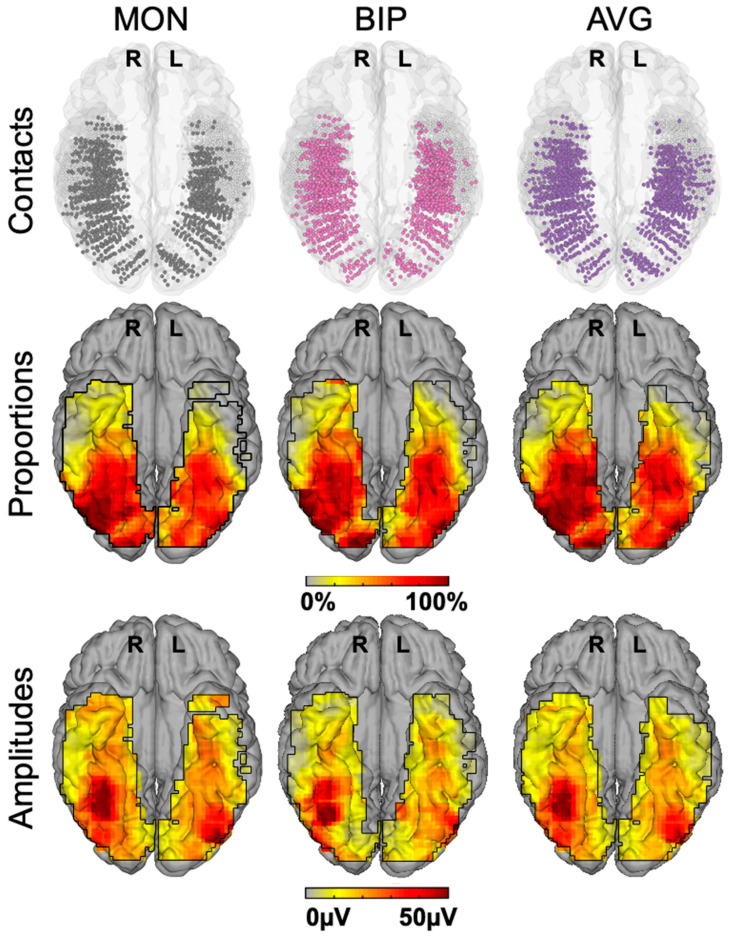
Face-selective maps are largely similar across reference montages (Hagen et al., in preparation) (MON: scalp electrode or intracerebral white-matter contact, BIP: bipolar montage; AVG: common average reference). **Top**: maps of all 5268 VOTC recording contacts across the 93 individual brains displayed in the Talairach space using a transparent reconstructed cortical surface of the Colin27 brain. Each circle represents a single contact, with colored-filled circles corresponding to selective contacts and gray-filled circles corresponding to non-selective contacts. **Middle** and **bottom**: Proportion and amplitude maps. Black contours outline proportions and amplitude significantly above zero.

**Figure 20 brainsci-13-00354-f020:**
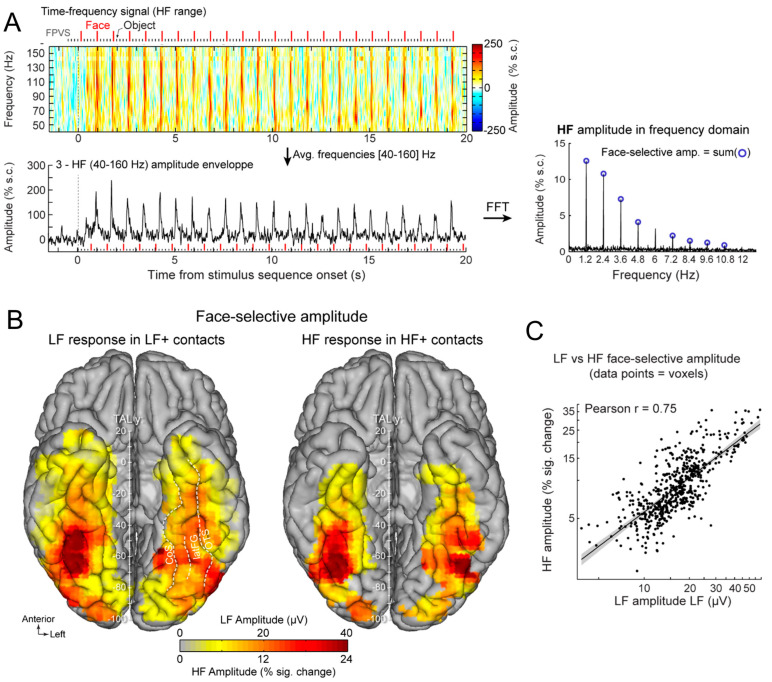
Recording and quantifying SEEG low frequency and HF face-selective signals in the VOTC (from [37]). (**A**) Top: The SEEG signal varying in time and frequency (40–160 Hz, using a Morlet wavelet approach) is shown from −1.5 to 20 s relative to the onset of an FPVS face-localizer sequence (see Figure 2A). The recording made from a contact in the latFG highlights a distinct periodic burst of HFB activity occurring around each face presentation, i.e., at the frequency of face stimulation (1.2 Hz). Bottom: The modulation of HFB amplitude over time (i.e., HF amplitude envelope) is obtained by averaging time-frequency signals across the 40–160 Hz frequency range. HFB face-selective amplitude is quantified by transforming the time-domain HF amplitude envelope to the frequency domain (Fast Fourier Transform, FFT) and summing amplitudes of the signal at 12 harmonics of the frequency of face stimulation (1.2, 2.4, 3.6, 4.8, … Hz, excluding harmonics of the 6 Hz base stimulation rate). (**B**) Maps showing smoothed low-frequency (left) and HF (right) face-selective amplitude displayed over the VOTC cortical surface. (**C**) A linear relationship between low-frequency and HFB amplitude maps is shown in panel (**B**). Each data point shows the face-selective amplitude in 12 mm × 12 mm voxels in Talairach space. Amplitudes were normalized using log transformation prior to computing the Pearson correlation. Only voxels overlapping across the two maps are used to estimate the Pearson correlation. The shaded area shows the 95% confidence interval of the linear regression line computed by resampling data points with replacement 1000 times.

**Figure 21 brainsci-13-00354-f021:**
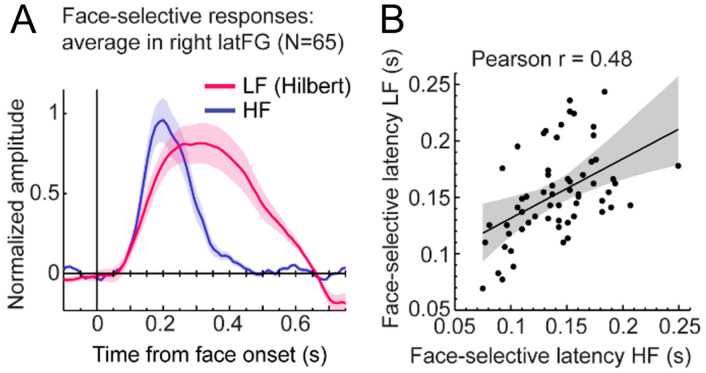
LF and HF timing relationship in the right latFG. (**A**) Time-domain face-selective responses for LF and HF averaged across 65 recording contacts in the right latFG. The shaded area shows the standard error of the mean across contacts. For LF, to limit the influence of variation in response morphology or polarity across recording contacts, a Hilbert transform was applied to the response of each contact before averaging. Averaged time-domain responses were then normalized (0 to 1) separately for LF and HF and aligned for their pre-face-onset amplitude level (−0.166 to 0 s). (**B**) Scatter plot showing the relationship between the onset latency of LF and HF face-selective responses measured in individual recording contacts in the right latFG. The shaded area shows the 99% confidence interval of the linear regression line computed by resampling data points with replacement 1000 times.

**Figure 22 brainsci-13-00354-f022:**
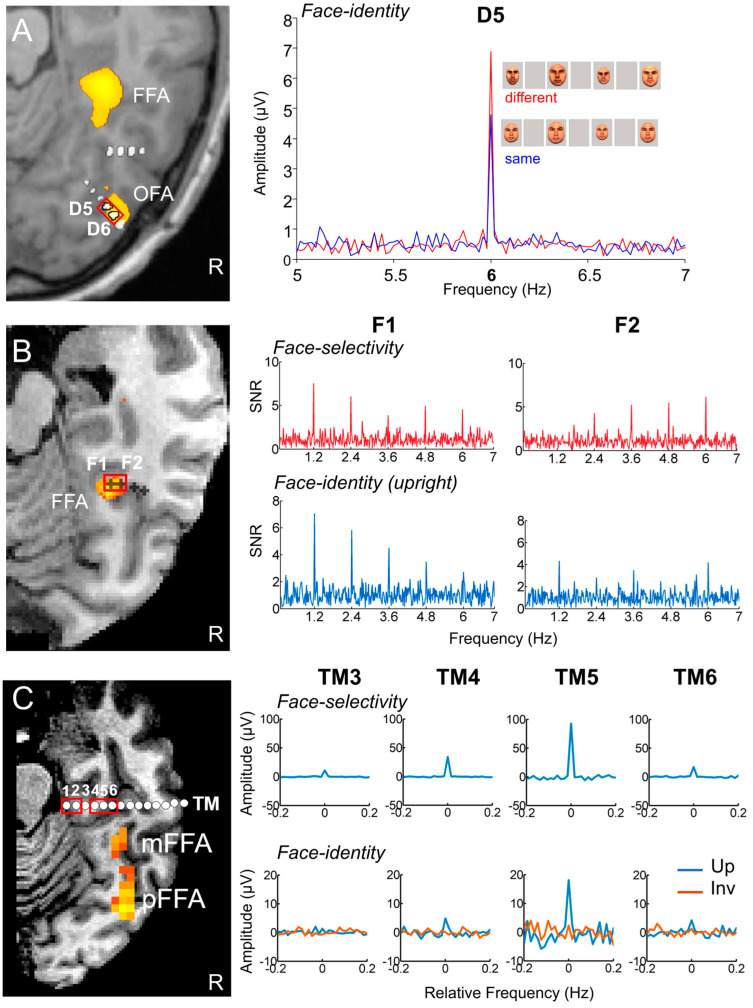
Direct electrical stimulation effects on FIR (transient prosopagnosia) are closely linked to local face-selectivity and sensitivity to face identity recorded on critical contacts. For all cases, the left panel shows the fMRI face-selective activations in the right VOTC (axial slices) with the SEEG electrodes superimposed (white dots, highlighted with red rectangles for critical contacts), and the right panel shows SEEG recordings during FPVS paradigms measuring face-selectivity (see paradigm in Figure 2A) and sensitivity to face identity. (**A**) Stimulating the right face-selective IOG (D5-D6) evoked a transient inability to discriminate unfamiliar face identities [248]. During SEEG, KV was tested with an FPVS adaptation paradigm measuring sensitivity to face identity at a fast rate of 6 Hz, with either identical faces or different faces [55]. The significantly largest difference for different versus same faces for upright faces, as well as the largest face inversion effect (i.e., upright-inverted), were found on the critical contact D5. (**B**) Stimulating the right face-selective LatFG (F1–F2) evoked a transient face identity palinopsia (subject MB, [233]). Of all the 137 recorded contacts in MB’s brain, contact F1 recorded, by far, the largest face-selective response amplitude and face identity discrimination response amplitude (see paradigm in Figure 17A). (**C**) Stimulating the right face-selective anterior fusiform gyrus (contacts TM1 to TM2 and TM4 to TM6) induced a transient inability to point out the familiar faces among unfamiliar faces and to match the identity of either familiar or unfamiliar faces (Subject DN, [97]). No fMRI face-selective activations were found in the vicinity of the stimulation sites because of a severe signal drop-out affecting this region. Of all the 141 recorded contacts in DN’s brain, one of the critical contacts (TM5) recorded the largest face-selective response amplitude and face identity discrimination response amplitude in the upright condition as well as the largest face inversion effect, i.e., upright-inverted (see paradigm in Figure 16A).

**Table 1 brainsci-13-00354-t001:** Advantages and weaknesses of intracerebral recordings with SEEG. Note that these characteristics are similar to ECoG.

SEEG Advantages	SEEG Weaknesses
Recordings in humans (in vivo) Direct recording of neural activity (i.e., not a metabolic correlate) Inside grey matter, including sulci (SEEG) High spatial and temporal resolution Possibility of focal electrical stimulation No major SNR fluctuations (i.e., signal drop due to artifacts) Complex datasets varying in space/time/frequency	Invasive Rare (typically small N) Potential brain lesions or malformations in epileptic patients, long-term epileptic seizures Integrity of brain function in the patients Few electrode locations (limited sampling) Electrode locations based on clinical purposes Variability across patients in electrode positions Complex datasets varying in space/time/frequency

## Data Availability

Not applicable.

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
