# Peer review of "Intracerebral Electrophysiological Recordings to Understand the Neural Basis of Human Face Recognition"

_brainsci, 2023, doi:10.3390/brainsci13020354_

Round 1

Reviewer 1 Report

This review related to the neural basis for face recognition using intracerebral recordings is a great work, very clear and very well structured. 

I only have a few minor comments:

- Line114-117: there are several types of intracerebral electrodes, each one with specific measurements. I propose the authors to mention that the measurements mentioned in the text are just an example of the ones they are using.

- A part of Figure 2 is missing/not visible,

- Lines 236-252: following the same rationale, how do the authors explain that during electrical stimulation on SEEG electrodes only a very limited area in the fusiform gyrus was able to determine prosopagnosia? The electrical stimulation follows the same rules of cortical spreading as spontaneous activity.

- Figure 11A – typo in the figure (House-selective)

- Figure 17A is truncated.

- Besides the epileptogenic zone and the epileptogenic lesion, there is also the irritative zone (brain region expressing spiking) and antiseizure medication load. Could the authors also address these issues in the limitation paragraph?

Reviewer 2 Report

This review does a great job in explaining the progress that has been made using intracranial EEG recordings in understanding neural processing of faces. Here are a few suggestions to help improve the manuscript.

1) Figure 6 caption page 12 – add “right panel” to this sentence to clarify as follows

“Amplitude across contacts can also be smoothed and projected on the cort2)ical sur-

face by plotting the mean face-selective amplitude within 15 x 15 mm voxels (right panel). “

2) There is some discussion of visual evoked potentials but the processing done in this manuscript does not involve signal averaging as is done in typical VEP processing. Is it possible to show averaged face evoked potential with signal averaging to understand the time course of neural activity during face processing?

3) There are several good comparisons and figures shown in this manuscript comparing the results between iEEG and fMRI. However, these techniques are measuring different physiological aspects of neuronal processing and it would helpful to show a table of strengths and weaknesses similar to table 1 but with iEEG and fMRI.
